# A Dynamic molecular basis for malfunction in disease mutants of p97/VCP

**Anne K Schuetz[1,2,3]\*, Lewis E Kay[1,2,3,4]\***

[1]Department of Molecular Genetics, University of Toronto, Toronto, Canada; [2]Department of Biochemistry, University of Toronto, Toronto, Cananda; [3]Department of Chemistry, University of Toronto, Toronto, Canada; [4]Program in Molecular Structure and Function, Hospital for Sick Children, Toronto, Canada

**Abstract** p97/VCP is an essential, abundant AAA+ ATPase that is conserved throughout eukaryotes, with central functions in diverse processes ranging from protein degradation to DNA damage repair and membrane fusion. p97 has been implicated in the etiology of degenerative diseases and in cancer. Using Nuclear Magnetic Resonance spectroscopy we reveal how disease-causing mutations in p97 deregulate dynamics of the N-terminal domain that binds adaptor proteins involved in controlling p97 function. Our results provide a molecular basis for understanding how malfunction occurs whereby mutations shift the ADP-bound form of the enzyme towards an ATP-like state in a manner that correlates with disease severity. This deregulation interferes with the two-pronged binding of an adaptor that affects p97 function in lysosomal degradation of substrates. Subtle structural changes propagate from mutation sites to regions distal in space, defining allosteric networks that facilitate inter-domain communication, with potential implications for modulation of enzyme activity by drug molecules.

\*For correspondence: anns@pound.med.utoronto.ca (AKS); kay@pound.med.utoronto.ca (LEK)

**Competing interests:** The authors declare that no competing interests exist.

## Introduction

The valosin containing protein (VCP) or p97 is a highly conserved enzyme in mammalian cells, with orthologues in yeast (Cdc48), in flies (TER94) and in archaea (VAT) (*Rabouille et al., 1995*). It is involved in various processes in the cell (*Figure 1A*) including membrane fusion (*Rabouille et al., 1995*), chromatin-associated functions (*Moreno et al., 2014*; *Dantuma and Hoppe, 2012*), cell cycle progression (*Cao et al., 2003*), and apoptosis (*Madeo et al., 1997*) and it is active in proteasomal degradation (*Richly et al., 2005*), autophagy (*Ju et al., 2009*; *Buchan et al., 2013*), and in endosomal pathways (*Ritz et al., 2011*). The involvement of p97 in all major proteolysis pathways makes it a central player in cellular homeostasis (*Meyer et al., 2012*). p97 is a 540 kDa homo-hexamer (6x89 kDa) with each monomer comprising an N-terminal domain (NTD) and a pair of ATPase domains, D1 and D2, arranged in primary sequence as NTD-D1-D2. Both D1 and D2 are organized as rings that stack coaxially, with the NTD located at the periphery of the D1 ring (*DeLaBarre and Brunger, 2003*) (*Figure 1B,C*). p97 converts the energy obtained via ATP hydrolysis to remove substrates from complexes or membranes and to structurally remodel or unfold them (*Barthelme and Sauer, 2016*). Its diverse activities result from various cofactors that recruit it to specific functions (*Buchberger et al., 2015*), with more than 40 adaptors discovered in mammalian cells. Many of these contain highly conserved p97 binding domains such as UBX (*Buchberger et al., 2001*) and VIM (*Stapf et al., 2011*). Given its prominent role in the eukaryotic cell, p97 misfunction is associated with human disease. While p97 deletion results in early embryonic lethality (*Müller et al., 2007*), a series of missense mutations lead to very specific malfunctions in protein homeostasis

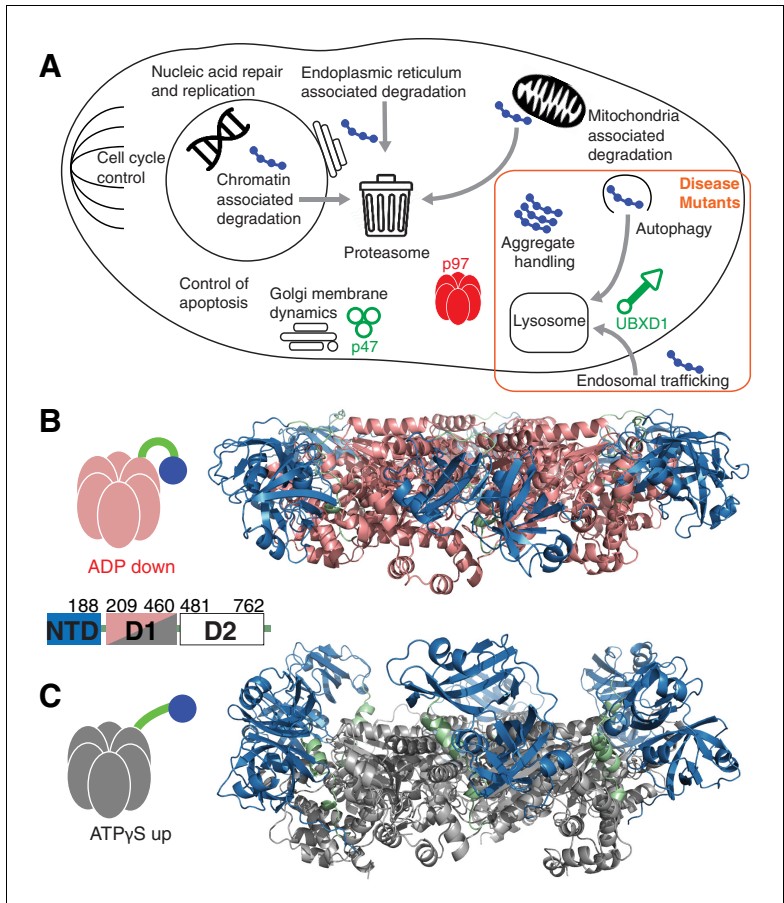

**Figure 1.** p97 structure and function. (**A**) Schematic illustration of a set of p97 cellular functions adapted from (*Meyer and Weihl, 2014*) highlighting pathways (orange square) that may be affected by IBMPFD disease-related mutations. Substrates are shown in blue and adaptors studied herein in green. (**B,C**) Ribbon-diagram representation of ND1Lp97 (residues 1–480) [PDB: 1E32 for ADP state, 4KO8 for ATPγS state of R155H] with NTD (blue), linker (green) and D1 (red in ADP state, **B**; grey in ATPγS state, **C**) color-coded. Shown to the side are cartoons of ND1Lp97-ADP and ND1Lp97-ATP that are used in other figures, with a single NTD highlighted in the down (ADP) and up (ATP) position, as well as the domain structure of p97.

The following figure supplement is available for figure 1:

**Figure supplement 1.** Impact of nucleotide on the structure of ND1Lp97.

linked to degenerative disorders (*Watts et al., 2004*), among them Inclusion Body Myopathy associated with Paget disease of the bone and Frontotemporal Dementia (IBMPFD), a lethal autosomal dominant disorder with onset in midlife. These mutations, mostly occurring at the NTD-D1 interface or in the linker region between these domains (*Figure 2*), also lead to an increased occurrence of amyotrophic lateral sclerosis (*Johnson et al., 2011*) and of familial Parkinson's disease (*Chan et al., 2012*).

Of particular functional relevance is the mobility of the NTD, which has been shown to undergo large displacements during the nucleotide cycle (*Tang et al., 2010*). NTDs are the major binding sites for cofactors along with substrates and cross-linking NTD to D1 via disulfides abrogates all ATPase activity (both in D1 and in D2) (*Niwa et al., 2012*). Crystallographic studies (*Tang et al., 2010*) conducted on a truncated form of p97 comprising NTD and D1 show that NTD is coplanar relative to D1 in both wild-type (wt) and disease mutants of p97-ADP (referred to in what follows as the down state, *Figure 1B*, *Figure 1—figure supplement 1A,B*) and elevated relative to D1 in p97-ATPγS disease mutants (up state, *Figure 1C*, *Figure 1—figure supplement 1A,B*). Complementary

**Figure 2.** IBMPFD mutation sites. (**A**) Residues that are mutated in IBMPFD patients and investigated in this study are indicated in yellow and ADP (grey) is shown with a space-filling model. (**B**) Table summarizing the distribution of 20 IBMPFD disease mutation sites across three NTD/D1 interfaces and the linker.

static pictures recently obtained from electron microscopy on full-length wt p97 (22) confirm that the nucleotide state in D1, but not in D2, determines the NTD up/down conformation and establish that the up-conformation in the ATPγS state is not an anomaly of the disease mutants but also found in wt protein. Thus, a picture emerges whereby NTDs of wt or mutant p97 exist in one of two conformations, controlled by the nucleotide state of D1 and not by mutation. A structural understanding of how disease mutations deregulate the ATP cycle and modulate p97 function is, therefore, lacking from the available static models. Herein this is addressed by using methyl-TROSY Nuclear Magnetic Resonance (NMR) spectroscopy, that is sensitive to protein dynamics, to show that IBMPFD disease mutations deregulate the up/down NTD equilibrium, leading to impaired binding of an adaptor involved in the lysosomal degradation pathway.

## Results

### Methyl-TROSY NMR of p97

We have used methyl-TROSY NMR spectroscopy that is optimized for studies of high molecular weight complexes (*Tugarinov et al., 2003*) to study a 320 kDa construct of p97 containing the NTD, D1 and the linker between D1-D2 (ND1Lp97, 6*53 kDa, residues 1–480) that has been used previously for crystallographic studies (*Tang et al., 2010*). Samples of highly deuterated, Iδ1-$^{13}$CH$_3$, *proR* L,V-$^{13}$CH$_3$, Mε-$^{13}$CH$_3$ (referred to as ILVM-$^{13}$CH$_3$-) p97 have been prepared following standard protocols (*Tugarinov and Kay, 2004*; *Gelis et al., 2007*), with methyl groups exploited as probes of molecular structure and dynamics. A high level of deuteration is required to improve spectral sensitivity and resolution by minimizing peak broadening that results from $^1$H-$^1$H spin relaxation interactions that would otherwise dominate in protonated samples of high molecular weight complexes (*Tugarinov et al., 2003*; *Sprangers and Kay, 2007*).

Well-resolved resonances in $^{13}$C-$^1$H HMQC spectra of ILVM-$^{13}$CH$_3$-ND1Lp97 and full length p97 (6*89 kDa) labeled in the same manner are superimposable (*Figure 3—figure supplement 1*), establishing that ND1Lp97 is a good model system for structural studies. Notably, some peaks in spectra of full-length p97 are missing in the comparison that reflects the slower tumbling of the larger complex, leading to inferior spectra relative to the 320 kDa construct. Substantial changes in spectra are noted for ND1Lp97 as a function of nucleotide (ADP or ATPγS) that are detected across the entire protein, consistent with a major rearrangement of the NTD from up to down as ATP is converted to ADP (*Tang et al., 2010*; *Banerjee et al., 2016*), *Figure 3*, *Figure 3—figure supplement 2*. Having established that high quality NMR spectra of ND1Lp97 could be recorded, we next assigned methyl cross-peaks to specific sites using a combination of mutagenesis and nuclear Overhauser effect spectroscopy (*Wüthrich, 1986*), taking advantage of the X-ray structures of this construct, as described previously (*Sprangers and Kay, 2007*; *Ruschak and Kay, 2012*). Approximately 97% and 79% of ILVM-methyl assignments in ADP and ATPγS states, respectively, were obtained in this manner (*Figure 3—figure supplement 3*).

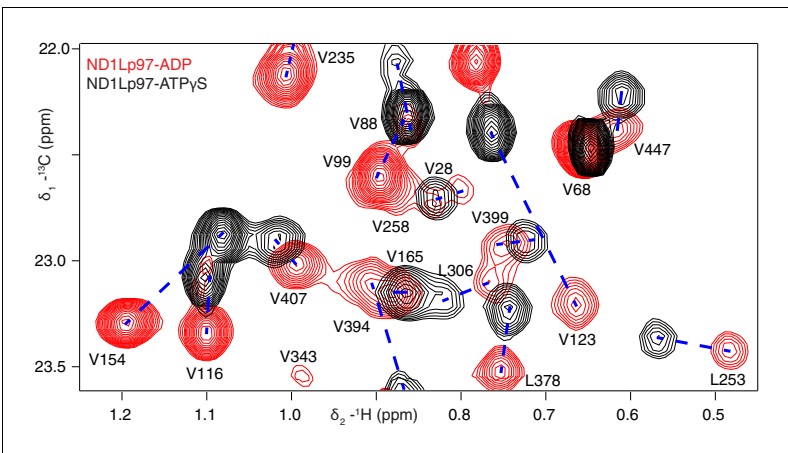

**Figure 3.** Nucleotide-induced structural changes in p97 as established by NMR. Superposition of a representative region from methyl-TROSY HMQC spectra of ILVM-$^{13}$CH$_3$-ND1Lp97 recorded at 800 MHz, 50°C. Dashed lines illustrate peak movement between nucleotide states. Selected methyl assignments have been included.

The following figure supplements are available for figure 3:

**Figure supplement 1.** ND1Lp97 is a good structural model for NTD and D1 of full-length p97.

**Figure supplement 2.** Nucleotide-induced ND1Lp97 spectral changes.

**Figure supplement 3.** Near complete methyl assignments for ND1Lp97-ADP.

## Disease mutations shift NTD equilibrium in p97 ADP state

Out of more than 20 different disease sites that have been reported to date in humans (*Darvish et al., 2004*; *Mehta et al., 2013*; *Evangelista et al., 2016*), 7 have been chosen for NMR analysis, *Figure 2A*. These include NTD residues R95 and R155 at the NTD-D1 interface, R191 and L198 in the NTD-D1 linker and the NTD-D1 interfacial residues A232, T262 and N387 in D1. A number of different mutants at position 155 were included (R155H, R155C and R155P), along with pairs of opposing residues at the NTD-D1 interface (R95G/T262A, R155H/N387H). The selected sites span the complete NTD-D1 interface (~7800 Å$^2$) and NTD-D1 linker region over which the disease mutations are localized. Moreover, the sites chosen for analysis are very close in space to the majority of the other disease causing mutations and thus are likely to be good reporters for these as well, *Figure 2B*.

The ATPγS state, ND1Lp97-ATPγS, is little affected by the mutations considered, beyond the immediate site of substitution, as can be seen in a comparison of spectra recorded on wt and R95G ND1Lp97 samples (*Figure 4A*, *Figure 4—figure supplement 1A*). Because chemical shifts are sensitive probes of structure this result strongly suggests that the disease mutants sample similar conformations as the wt protein in the ATPγS-bound form. In contrast, substantial changes are observed for many methyl probes in ND1Lp97-ADP, including those in the NTD, N-D1 linker and D1 regions. These changes in chemical shifts generally titrate with mutation in a linear fashion, (*Figure 4B*, *Figure 4—figure supplement 1B*), not withstanding the methyl groups close to the sites of mutation or nucleotide binding whose positions can be perturbed simply by the substitution or the presence of ADP/ATP, respectively. Deviations from linearity, observed for a fraction of the probes, likely reflect small structural perturbations beyond the principal effect of the mutation (see below). In this context, it is noteworthy that in the general case non-linear changes in shifts can be observed even in the case of two-site exchange when the chemical shift time-scales for $^{13}$C and $^1$H nuclei are different, that can arise from large chemical shift differences for one nuclei and smaller changes for the other. Thus, although mutations are localized to different domains and can be distant in space from each other, their impact on the structure and dynamics of ND1Lp97-ADP can be described in terms of a simple, unifying model whereby mutations lead to a gradual conversion to a more ATP-like

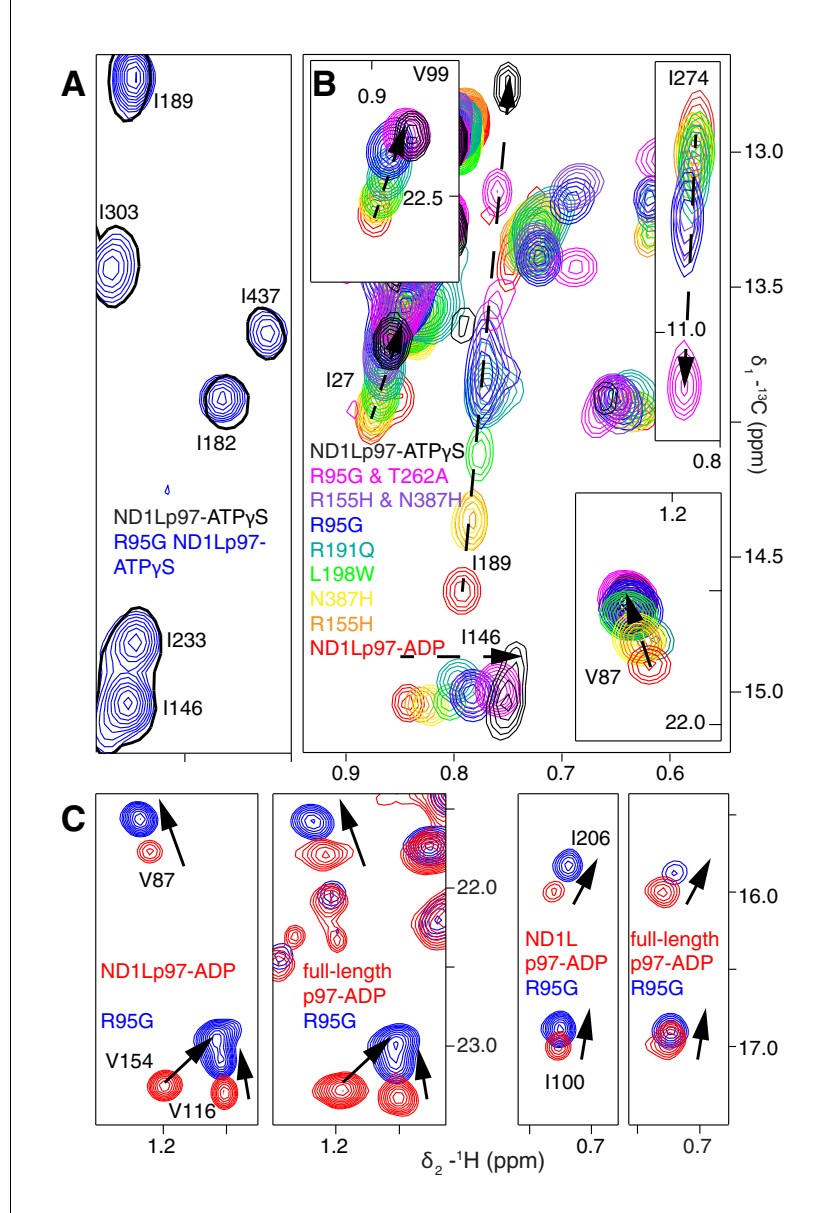

**Figure 4.** A dynamic NTD equilibrium between up/down states is affected by mutations in ND1Lp97-ADP.
(A) Similar spectra are obtained for wt (black, single contour) and R95G (blue, multiple contours) ND1Lp97-ATPγS.
(B) Superposition of selected $^{13}$C-$^1$H HMQC spectral regions of wt (red) and mutant ND1Lp97-ADP (colored as indicated) and of wt ND1Lp97-ATPγS (black), focusing on V87, V99, I146 from the NTD, I189 from the NTD-D1 linker and I274 (D1), showing the progressive titration of cross-peak chemical shifts. Note that the I274 peak for wt ND1Lp97-ATPγS is not shown; a large CSP is noted from the ADP to ATPγS substitution due to the proximity of I274 to the nucleotide. (C) Superposition of selected regions of $^{13}$C-$^1$H HMQC spectra of full-length wt and R95G p97 (ADP state, right) showing analogous changes as for ND1Lp97-ADP (left).

The following figure supplement is available for figure 4:

**Figure supplement 1.** Disease mutation-induced ND1Lp97 spectral changes.

conformation (red to black in *Figure 4B*, despite the absence of ATP), consistent with a gradual upward movement of the NTD. This conversion is readily apparent for I189, for example, located in the NTD-D1 linker, that is sensitive to a loop to helix conformational change that accompanies the down to up NTD conformation and has the largest chemical shift perturbation (CSP) between ADP- (red) and ATPγS- (black) bound states. The chemical shifts in each of these states provide the limiting values from which the fraction of NTDs that are up/down can be readily calculated for the different mutants, as described below.

A concern in studies of smaller fragments of intact proteins is the possibility that the absence of part of the structure may bias the results. As a control we have also studied full-length p97. *Figure 4C* shows the superposition of regions of $^{13}$C-$^{1}$H HMQC spectra recorded of both wt and R95G variants of ND1Lp97-ADP and full-length p97-ADP. It is clear that the CSPs for both ND1Lp97 and full-length p97 are identical so that the upwards movement of the NTD upon introduction of disease mutations, characterized in ND1Lp97-ADP, is also observed in the intact molecule. Thus, unlike the picture from X-ray studies of disease mutants, our NMR results establish that the NTD is not static in p97-ADP, but rather exchanges rapidly between up/down states, with the relative mixture of each state changing with mutation. The exchange rate is estimated to be greater than 2000 s$^{-1}$, based on the chemical shift differences between ADP and ATPγS-bound states, as measured for the I189 Cδ1 that has the largest CSPs. Moreover, less severe mutations (see below), such as R155H, have a relatively small effect on the NTD equilibrium (orange peaks in *Figure 4B*), which is skewed to the down state, but the severe R95G mutation (blue) shifts the relative up/down populations to approximately equal.

*Figure 5A* shows all ILVM methyl groups that have been used as probes superimposed on the structure of ND1Lp97, with each methyl indicated by a sphere that is color-coded by the difference in chemical shifts in spectra recorded of R95G ND1Lp97-ADP and wt ND1Lp97-ADP constructs. CSPs extend from the site of mutation (yellow) to the nucleotide-binding region and beyond to the p97 central pore, identifying potential allosteric pathways of communication throughout the protein (see Discussion). The 22 residues in ND1Lp97 that are mutated in IBMPFD disease patients are shown in stick representation in red in *Figure 5B*. These mutation sites localize to regions with large CSPs. Notably, regions of the NTD that are associated with UBX and VIM domain binding (see below) are not perturbed by the disease mutations (no CSPs).

## Disease mutations decrease interactions at the NTD-D1 interface

Further insight into the NTD equilibrium can be obtained from experiments that measure the overall tumbling time, $\tau_c$, of the NTD in the context of the ND1Lp97 hexamer in different nucleotide states and for different mutations. We have obtained NTD $\tau_c$ values by quantifying the build-up of methyl $^{1}$H triple-quantum coherence that occurs efficiently only for high molecular weight complexes and rigid methyl probes (*Sun et al., 2011*) (see Materials and methods for details, *Figure 6—figure supplement 1*). *Figure 6A* illustrates a number of build-up profiles focusing on examples from p97 constructs at different ends of the NTD dynamics spectrum. In the limit of a rigidly attached NTD $\tau_c$ is predicted to be approximately 120 ns (as detailed in Materials and methods), while for a fully independent domain a value of approximately 13 ns is expected. Correlation times of 91 ± 3 ns and 68 ± 2 ns were determined for ADP and ATPγS bound wt states, respectively, while a value of 13 ns was measured for the isolated NTD. This implies that the NTD is only partially docked to D1, as expected on the basis of X-ray and cryo-EM models of p97 (20, 22), with more motional freedom in p97-ATPγS. A value of $\tau_c$ = 100 ± 3 ns was obtained for an intra-protomer cross-linked variant that tethers the NTD to D1 (*Figure 6A*).

To quantify how the NTD tumbling time is correlated with the up/down equilibrium we have calculated the fractional population of NTD in the up state, $p_U$, for each of the variants examined. Since plots of chemical shifts *vs* mutation are linear with little broadening during the course of the chemical shift *vs* mutation trajectory (*Figure 4B*) it is reasonable to assume a two-site exchange mechanism between up/down NTD states that is fast on the NMR chemical shift time-scale. In this limit, $p_U$ values can be calculated according to the relation (*Palmer et al., 2001*)

$$\delta_{mutant} = p_D \, \delta_D + p_U \, \delta_U \qquad (1)$$

where $\delta_{mutant}$ is the chemical shift of a methyl probe in the mutant, $\delta_D$ and $\delta_U$ are the corresponding

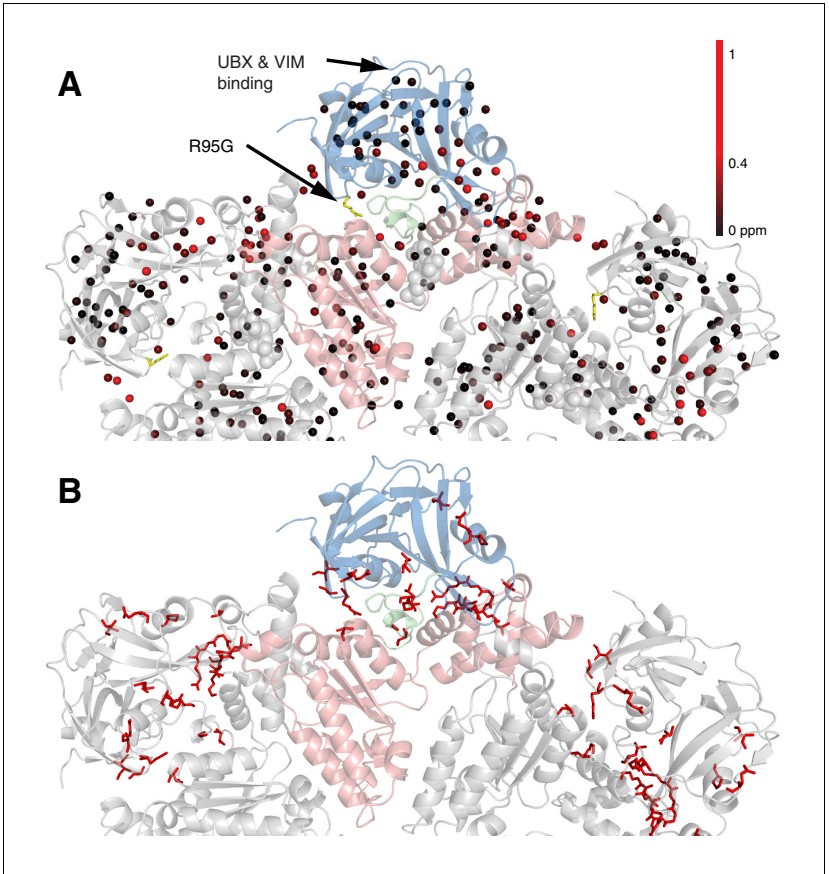

**Figure 5.** Methyl CSPs introduced by the R95G mutation. (**A**) Methyl groups are shown as spheres and color-coded according to the size of the CSP, as indicated. The UBX/VIM (*Buchberger et al., 2001*; *Stapf et al., 2011*) binding groove between the two NTD subdomains is unaffected by disease mutations. (**B**) For reference, the locations of the 22 identified sites of IBMPFD mutations in p97 ND1L are shown in red.

chemical shifts in the ADP (down) and ATPγS (up) states and $p_{D/U}$ is the fractional population of the $D/U$ state ($p_U + p_D = 1$). Values of $p_U$ for a range of disease mutants are listed in *Figure 6B* (purple) along with $\tau_c$ values (blue), with $\tau_c$ decreasing (mobility increasing) as the percentage of NTD in the up conformation grows, *Figure 6C*. It is worth noting that I189, L229 and M388 all have large CSPs due to mutation and the $p_U$ values obtained from the trajectories of all 3 residues are consistent (see legend to *Figure 4—figure supplement 1B*).

Both the chemical shift and relaxation data provide strong evidence that the NTDs are dynamic, rapidly interconverting between up/down conformers. In further support of this interpretation we have carried out chemical cross-linking experiments by introducing cysteine residues at positions 155 (NTD) and 387 (D1) that are within 3 Å in the down state (*Niwa et al., 2012*), yet greater than 20 Å in the up conformation. The R95G mutant has been chosen for cross-linking because it is effectively in a '50% up' conformation (*Figure 6B*) so that in the limit of a static structure a disulfide bond connecting C155 and C387 would not be formed and CSPs would not be observed upon oxidation of the sample. The $^{13}$C-$^1$H HMQC spectrum of R95G,R155C,N387C ND1Lp97-ADP, *Figure 6D*, under oxidizing conditions shows large CSPs for I146 and I189, consistent with disulfide bond formation. The presence of a disulfide bond is further established by comparing the chemical shifts of these peaks (grey) with those for R155C,N387C ND1Lp97-ADP upon crosslinking, *Figure 6E* (grey), where a locked, down NTD orientation is readily formed upon oxidation. Taken together our data indicate that the disease mutants affect the up/down equilibrium in p97-ADP, with the NTD almost entirely up in the R95G/T262A double mutant, similar to the ATPγS-conformation (*Figure 6B*). Finally, it is worth noting that the positions of the I146 and I189 cross peaks under reducing

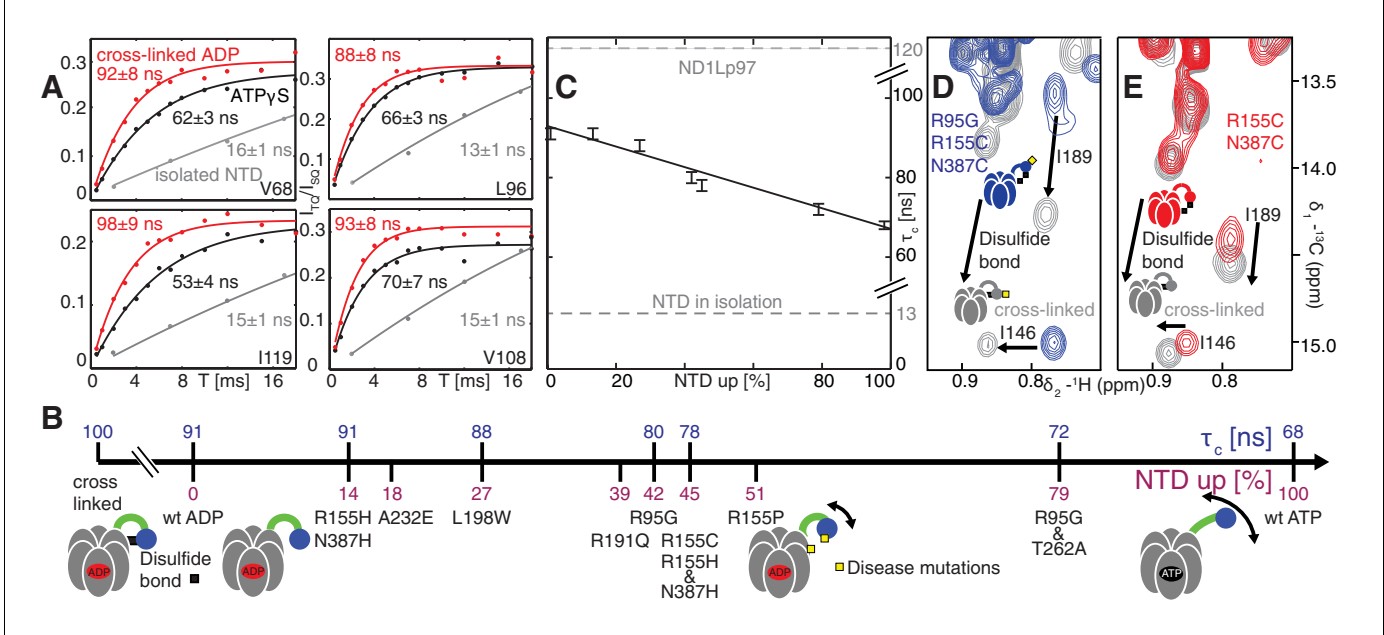

**Figure 6.** Probing NTD dynamics in ND1Lp97-ADP. (A) Measurement of NTD tumbling times for ND1Lp97-ADP cross-linked to the down position via a disulfide link between positions 155 (NTD) and 387 (D1) (320 kDa, 50°C, red), ND1Lp97-ATPγS (NTD up, 50°C, black) and isolated NTD (24 kDa, 37°C, grey). Correlation times were obtained via an approach that monitors the build-up of triple quantum coherence (see Materials and methods). (B) NTD tumbling times ($\tau_c$) and fraction NTD up ($p_U$), quantified by the $^{13}$C chemical shift of I189 for different mutants and nucleotide states of ND1Lp97. Increased motion corresponds to lower $\tau_c$. (C) Linear correlation of % NTD up vs NTD $\tau_c$ for different disease mutants. $\tau_c$ values for the isolated NTD (adjusted for 50°C) and ND1Lp97 (50°C), calculated from peaks in D1 (see Materials and methods), are given by dashed grey lines for reference. Chemical crosslinking via disulfide bond formation between cysteine residues at positions 155 and 387 (21) forces NTDs of both R95G ND1Lp97-ADP (D) and ND1Lp97-ADP (E, wt at position 95) to the down position, and increases $\tau_c$ to 100 ns in both cases.

The following figure supplement is available for figure 6:

**Figure supplement 1.** Histograms of $\tau_c$ distributions as obtained from per-residue fits of methyl $^1$H spin relaxation data.

conditions (*i.e.,* absence of disulfide bond) are very different for R95G,R155C,N387C ND1Lp97-ADP (blue) and R155C,N387C ND1Lp97-ADP (red) that reflects the different $p_U$ values for the NTDs, as discussed above.

## Implications of mutations on UBXD1 binding

We next sought to establish the consequences of the mutations that perturb the up/down equilibrium for adaptor binding. The UBXD1 adaptor protein is abundant in neuronal cells (*Madsen et al., 2008*; *Nagahama et al., 2009*) and a complex with p97 binds to the plasma membrane protein caveolin-1, mediating sorting of ubiquitylated cargo in endosomal degradation (*Ritz et al., 2011*) (*Figure 1A*, region highlighted in the orange square). Importantly, the UBXD1-p97 complex is specifically disrupted in disease-associated mutations and function is impaired (*Ritz et al., 2011*) but the molecular details underlying this disruption are not understood. An elegant biophysical study of the isolated NTD with the N-terminal portion of UBXD1 (UBXD1-N, residues 1–133, *Figure 7A*) (*Trusch et al., 2015*) shows an interaction involving VIM (residues 52–63) that is localized to the canonical VIM/UBX binding site on the NTD (*Stapf et al., 2011*). Moreover, the authors suggest that the N-terminus of UBXD1 may engage the NTD-D1 interface and NTD-D1 linker, where disease mutations occur, and hypothesize that this secondary interaction may favor a more compact conformation of p97. We have used methyl-TROSY NMR to study the binding of UBXD1-N to wt and mutant forms of ND1Lp97-ADP, *Figure 7*, *Figure 7—figure supplement 1–3*. In *Figure 7* the focus for simplicity is on three residues. First, V68 that is localized to the UBX/VIM binding region of the NTD and that, therefore, is a proxy for binding of VIM, and, second, I146 and I189 that report on

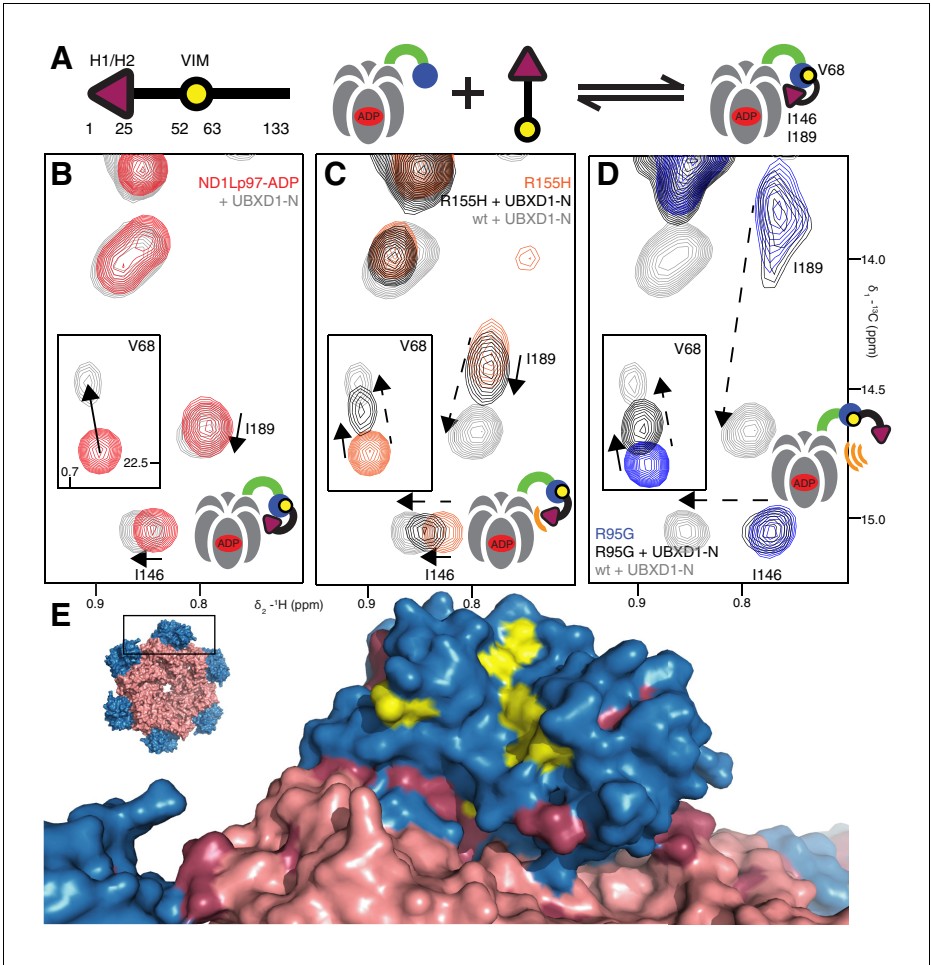

**Figure 7.** Disease mutants impair binding of UBXD1-N. (**A**) (left) Domain organization of UBXD1-N (residues 1- 133 (35)), (right) schematic of binding reaction highlighting 3 key residues that are used as probes of binding in what follows. (**B**) Selected regions of $^{13}$C-$^1$H HMQC spectra of wt ND1Lp97-ADP without (red) and with (grey) 3-fold excess UBXD1-N over protomer, focusing on V68, reporting on VIM domain binding, as well as I146 and I189 that serve as proxies for binding of the H1/H2 motif of UBXD1 (see text). As the NTD is in the down position prior to UBXD1-N binding, only small CPSs are observed for I146/I189, that reflect binding of H1/H2. (**C**) Addition of 3 fold excess UBXD1-N to R155H results in CSPs (orange to black) that do not extend to the fully bound state observed in wt (grey), indicating only a partial shifting of the up/down equilibrium to the down conformation (cartoon inset). Note that I146/I189 peaks for unbound R155H are shifted upfield relative to wt (compare orange with red peaks) reflecting the increased up population of the NTD for R155H (14%); differences in peak positions for wt and mutant p97 in other panels also reflect changes in the up/down equilibrium. (**D**) Addition of UBXD1-N to R95G ND1Lp97-ADP (42% up in the unbound state) results in partial binding of VIM but no binding of H1/H2 and subsequently no shift in the up/down equilibrium. (**E**) Chemical shift perturbations caused by binding of VIM (yellow) and H1/H2 (dark red) of UBXD1 to wt ND1Lp97-ADP mapped onto a surface representation of the NTD (blue) and neighboring D1 structure (inset shows complete hexameric structure from which the surface was taken).

The following figure supplements are available for figure 7:

**Figure supplement 1.** Positions of 3 key reporting residues, V68 (sensitive to VIM binding), I146 and I189 (both Ile are sensitive to the NTD up/down equilibrium) superimposed on a structural model of a complex of the VIM domain (yellow) and ND1Lp97-ADP.

**Figure supplement 2.** Adaptor binding of UBXD1 to wt and mutant p97.

**Figure supplement 3.** Titration of wt ND1Lp97-ADP with UBXD1-N to obtain an effective macroscopic $K_d(K_{d,macro})$ for the binding reaction.

binding of the H1/H2 region of UBXD1-N (*Figure 7*, *Figure 7—figure supplement 1*). Neither I146 nor I189 show CSPs from the addition of p47 UBX (*Figure 8—figure supplement 1A,B*), consistent with the fact that these residues are not sensitive to binding at the canonical UBX/VIM groove. Binding of UBXD1-N causes small shifts to I146 and I189 of the wt protein (*Figure 7B*) that are consistent with locking the NTD in the down conformation, as observed upon chemical cross-linking (*Figure 6E*). Larger shifts are observed for V68 and I175 that report on VIM binding, from which a $K_d$ value of $22 \pm 2$ μM for the VIM-NTD interaction is calculated (see Materials and methods *Figure 7—figure supplement 3*). This value is in good agreement with affinities measured for the binding of full-length wt p97 to full length UBXD1, 3.5 μM (*Hänzelmann and Schindelin, 2011*), and for the binding of the isolated NTD with UBXD1-N, 9 μM (*Trusch et al., 2015*), especially considering that the latter pair of studies were carried out at 25°C in comparison to 50°C for the NMR results reported here. Studies of R155H and N387H mutants of ND1Lp97-ADP that mildly perturb the up/down equilibrium (*Figure 7C*, *Figure 7—figure supplement 2D*) show that two-pronged binding involving VIM and H1/H2 occurs partially but that a complete down conformation is not obtained with a 3-fold excess of adaptor over p97 protomer (compare dashed and solid arrows that correspond to movements for peaks from wt and mutant p97, respectively). A $K_d$ value of ~80 μM is obtained for the VIM-NTD interaction in this class of mild disease mutants. Notably, while the VIM domain weakly binds to the NTD of the strong mutants R95G and R155P ND1Lp97-ADP, *Figure 7D* and *Figure 7—figure supplement 2E*, ($K_d \sim 200$ μM, see Materials and methods) there is no binding of H1/H2 and no shift in the conformational equilibrium towards the down state, as evidenced by a lack of CSPs for I146 and I189 (3:1 UBXD1-N:p97 monomer ratio). Supporting data from further probes are highlighted in *Figure 7—figure supplement 2*. These include V38 and V108 in the UBX/VIM binding site of NTD that are sensitive to the interaction with the VIM domain (*Stapf et al., 2011*), and V116, V133 and V154 reporting on H1/H2 binding. Finally, it is worth emphasizing that the similar trajectories of CSPs as a function of added UBXD1-N observed for wt and all disease mutants (note the collinearity of dashed and solid arrows) provides strong evidence that the mechanism of binding is identical in all cases, with the differences in the extent of shift changes reflecting differing affinities, as calculated in the present study.

Our results thus establish a link between the severity of the disease mutant and the ability of UBXD1 to shift the up/down equilibrium so as to lock the NTD in the down conformation. *Figure 7E* plots CSPs obtained from VIM (yellow) and from H1/H2 (purple) binding on a space filling model of the NTD, showing that the canonical VIM binding site is affected by the interaction with VIM (*Hänzelmann and Schindelin, 2011*), while the NTD-D1 interface serves as the binding site for H1/H2. A simple binding model for the UBXD1-N - p97 interaction is provided in the Materials and methods.

## Effects of disease mutations are more severe for specific adaptors

It is notable that IBMPFD disease mutants affect lysosome-related functions but not others, such as endoplasmic reticulum associated degradation and Golgi membrane fusion (*Tresse et al., 2010*; *Meyer and Weihl, 2014*; *Johnson et al., 2015*) (*Figure 1A*). We therefore sought to establish that the mutants considered here would not disturb complex formation with p47, one of the best-characterized adaptors and known inhibitors of p97, that is involved in regrowth of Golgi from membrane fragments (*Kondo et al., 1997*). As a first test we studied the p47 UBX domain exclusively, which bound to wt and R95G ND1Lp97-ADP identically (*Figure 8—figure supplement 1A,B*), in agreement with the literature (*Fernández-Sáiz and Buchberger, 2010*). Very similar CSPs were further obtained for wt and R95G proteins upon binding full-length p47 (*Figure 8A*, *Figure 8—figure supplement 1C*), which interacts as a trimer (*Beuron et al., 2006*). Notably, the CSPs upon binding p47 are distinct both in magnitude and in direction from those that result from perturbation to the NTP up/down equilibrium (*Figure 8A*, *Figure 8—figure supplement 1C*). For example, addition of p47 leads to shifts in I189 that are predominantly horizontal (solid arrows), compared with near vertical shifts (dashed arrows) that are observed as the NTDs become progressively detached from the D1 domain. Significant differences in CSPs are also observed between the I146 response to p47 binding and to changes in the up/down equilibrium as reported by I146 (compare dashed and solid arrows for I146). This suggests a different binding mechanism for p47 relative to UBXD1-N, that is made clear when methyl probes showing CSPs from p47 binding are highlighted on the ND1Lp97 structure, *Figure 8B*. Binding occurs at two sites, with one region corresponding to the canonical UBX

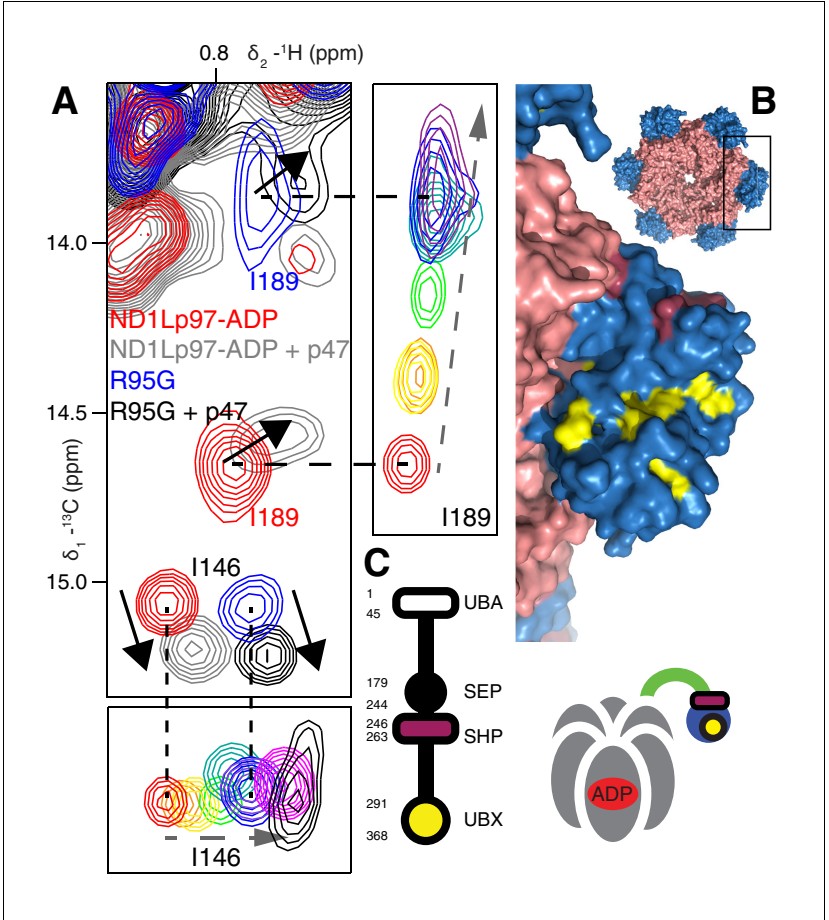

**Figure 8.** Binding of p47 is not impaired by disease mutations. (**A**) Superposition of selected regions of $^{13}$C-$^1$H HMQC spectra focusing on I146 and I189 of wt ND1Lp97-ADP and R95G ND1Lp97-ADP without (red for wt, blue for R95G) and with (grey for wt, black for R95G) 1.25 fold excess p47. Similar CSPs indicate that binding to p47 has not been impaired by the R95G mutation, nor is the up/down equilibrium changed. For reference, the 'mutant titration' from *Figure 4B* is provided at the sides of the main spectrum for residues I146 and I189, emphasizing that binding of p47 results in changes that are distinct from NTD up/down. (**B**) CSPs caused by binding of full-length p47 to wt ND1Lp97-ADP as mapped onto a surface representation of the NTD (blue) and neighboring D1 structure (light red), with the inset showing the complete hexameric structure from which the surface was taken. Residues affected by UBX binding are color-coded in yellow and those perturbed by SHP binding indicated in dark red. (**C**) Domain organization of p47 and cartoon of the p47-p97 complex, highlighting the regions of interaction.

The following figure supplement is available for figure 8:

**Figure supplement 1.** Adaptor binding of p47 to wt and mutant p97.

binding site of the NTD (yellow), as observed for the binding of the VIM domain of UBXD1-N. A second unique site on the NTD (purple) is also involved in binding that has been shown in a previous structural study to engage the SHP domain of p47 (*Hänzelmann et al., 2016*). Unlike the second site for UBXD1-N binding that bridges the NTD and D1 domains, the site of engagement of SHP does not, so that it is not surprising that the p47-p97 interaction is not significantly perturbed by disease mutations. *Figure 8C* illustrates schematically the binding of p47 SHP and UBX domains to p97.

## Discussion

The development of significantly improved NMR instrumentation, new labeling methodologies (*Tugarinov and Kay, 2004*; *Gelis et al., 2007*; *Kainosho et al., 2006*; *Gans et al., 2010*) and novel NMR experiments that exploit the labeling in ways that preserve signal intensity (*Tugarinov et al., 2003*; *Fiaux et al., 2002*) has significantly impacted on the size of protein complexes that are now amenable for detailed solution NMR spectroscopy studies (*Rosenzweig and Kay, 2014*). Here we have used experiments that rely on a methyl-TROSY effect to quantify inter-domain dynamics in a series of IBMPFD mutants of human p97 and to study interactions with adaptor molecules. Many studies of molecular machines by NMR have focused on homo-oligomeric systems where each protomer is relatively small (20–30 kDa) (*Sprangers and Kay, 2007*; *Shi and Kay, 2014*). In the case of ND1Lp97, that is the subject of the present work, protomeric molecular weights in excess of 50 kDa challenge resolution in spectra, necessitating the use of an elegant labeling strategy developed by Boisbouvier and coworkers whereby methyl groups are $^{13}CH_3$-labeled in a stereospecific manner (*Gans et al., 2010*). Near complete assignments for ILVM methyl groups (pro-R for LV) are reported for both ADP and ATP states of the enzyme (*Figure 3—figure supplement 3*).

To date over 40 different IBMPFD disease mutations have been identified in 22 different positions throughout ND1Lp97 (*Evangelista et al., 2016*), of which 7 representative sites were studied here. From our dynamics studies a picture emerges whereby the NTDs of p97-ADP interconvert between up/down conformations that can be modulated by IBMPFD disease mutations. X-ray and cryo-EM studies have established that for the wt protein the up/down NTD equilibrium is highly skewed to the down state in the ADP bound form (*Tang et al., 2010*; *Banerjee et al., 2016*; *Zhang et al., 2000*), and our NMR results show that this arrangement facilitates the two-pronged binding of UBXD1 to the enzyme. Our NMR studies further show that IBMPFD disease mutations perturb this equilibrium in the direction of the up state, leading to an impaired biological 'readout', namely UBXD1 adaptor binding.

A key strength of NMR in studies of proteins lies in the fact that the easiest parameter to measure – the chemical shift – is also amongst the most powerful in detecting subtle conformational changes of the sort that can give rise to allosteric pathways of communication, for example. Detailed crystallographic studies of p97 have established that the NTD/D1 interface and the conformation of the linker between NTD and D1 are coupled to the nucleotide state in D1 (20, 50). Notably, in the wt protein the nucleotide state in D1 determines the NTD up/down conformation and, conversely, perturbing the NTD down state by the introduction of disease mutations in the interface region propagates to the nucleotide-binding pocket (*Figure 5*). CSPs from the R95G mutation are indicated in *Figure 5A*; these provide some insight into the communication between distal regions of p97 at the level of individual amino acids, as highlighted in *Figure 9*. Here we show a number of pathways (**I-III**) defined by CSPs that arise upon perturbing the up/down equilibrium. A first pathway (**I**) includes residues in the linker connecting NTD with D1, with large CSPs observed for I189 and L198 at opposite ends of the linker; CSPs are also observed for I206 and L213 proximal to the nucleotide binding site. The arginine finger (R359) and Walker B (E305) residues, at the opposite face of the nucleotide binding site, sense the up/down equilibrium (in the case of the wt protein, ADP versus ATP) leading to CSPs close to the central pore, as observed for I274. Pathway **II** involves residues from an extensive interface between NTD and D1, including numerous disease mutation sites such as R155 and N387, and perturbations to the up/down equilibrium lead to changes far into the D1 domain. Finally, the NTD conformational change is communicated to the adjacent protomer at the I27/L429 interface via pathway **III** and this network eventually connects to the same cluster of residues as in pathway **II**.

Despite small patient groups and phenotypic variability it is possible to classify mutants as moderate or severe, based on age of onset and relative elevation of disease markers (*Figure 10A*), with R155H and R155C, the two most frequent mutations in patients (*Mehta et al., 2013*), defined as weak and strong, respectively. An important result from this study is the strong correlation between disease severity and the extent to which the up/down equilibrium is skewed in the ND1Lp97-ADP state with more severe mutants unable to engage the H1/H2 region of UBXD1 that is required to lock the NTD conformation in the down state, illustrated schematically in *Figure 10B*. Because a complex of UBXD1-p97 is required for recruitment of ubiquitylated caveolin-1 to the lysosome for degradation this pathway becomes impaired in the disease mutants, leading to the accumulation of caveolin-1 positive endolysosomes in IBMPFD patients (*Ritz et al., 2011*). Notably, significantly

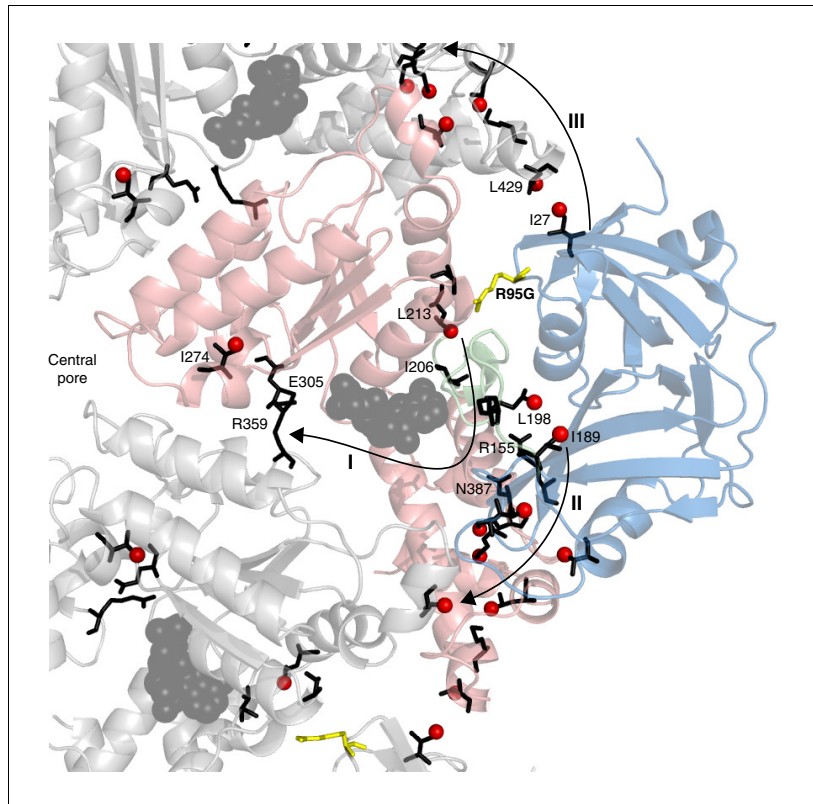

**Figure 9.** Putative allosteric networks illustrating 'pathways of communication' (I-III) from the site of the R95G disease mutation to regions distal in the structure. R95 is highlighted in yellow. Chemical shift perturbations upon R95G mutation above 0.3 ppm are shown as red spheres. Key residues that form continuous pathways are shown as black sticks, as discussed in the main text.

impaired binding to IBMPFD disease mutants of p97 is not observed for all adaptors. For example, we have shown that binding of p47 is not perturbed by mutation (see below), consistent with the location of observed CSPs showing that p47-p97 interactions only engage the NTD and do not include the NTD-D1 interface, as for UBXD1 (compare *Figure 7E* and *Figure 8B*).

It is of interest to ask why these structural changes have not been observed via other high-resolution techniques. The answer partly lies in the fact that the mild disease mutants introduce only subtle changes to the equilibrium, with R155H and N387H mutations shifting the NTD equilibrium from 0% up to ~15% up in p97-ADP, for example. Moreover the differences in free energies between the up and down conformations are relatively small, with $\triangle G_{up/down}$(R155H) ~ 1200 cal/mol and $\triangle G_{up/down}$(R95G) ~ 200 cal/mol based on the measured $p_U$ values (*Figure 6B*). The equilibrium may well be affected by the liquid nitrogen temperatures used in x-ray and cryo-EM studies and by crystal packing forces that could skew populations. The significant conformational dynamics associated with the up/down interconversion may also make it difficult to observe this effect by crystallography. For example, while crystal structures of weak disease mutants in the ADP state (R155H, L198W) have been published and all show the down NTD conformation, as for the wt, structures of strong mutants such as R95G, R155C/P in the ADP state ($\triangle G_{up/down}$~0 cal/mol) have not been reported. On an atomic scale, only localized changes were observed in the crystal structures of mutants of p97-ADP relative to the wt conformation (*Tang et al., 2010*; *Tang and Xia, 2013*; *Tang and Xia, 2016*). For example, the sidechain of R359, which serves as a sensor of the nucleotide state of an adjacent protomer, is slightly shifted and this shift is thought to lead to a destabilization of the ADP state in mutants. Moreover, the sidechain orientation of F360 differs between ADP states of wt and disease mutants (*Tang and Xia, 2013*). Together, these structural differences are thought to result in defective communication between subunits in mutant p97 molcules.

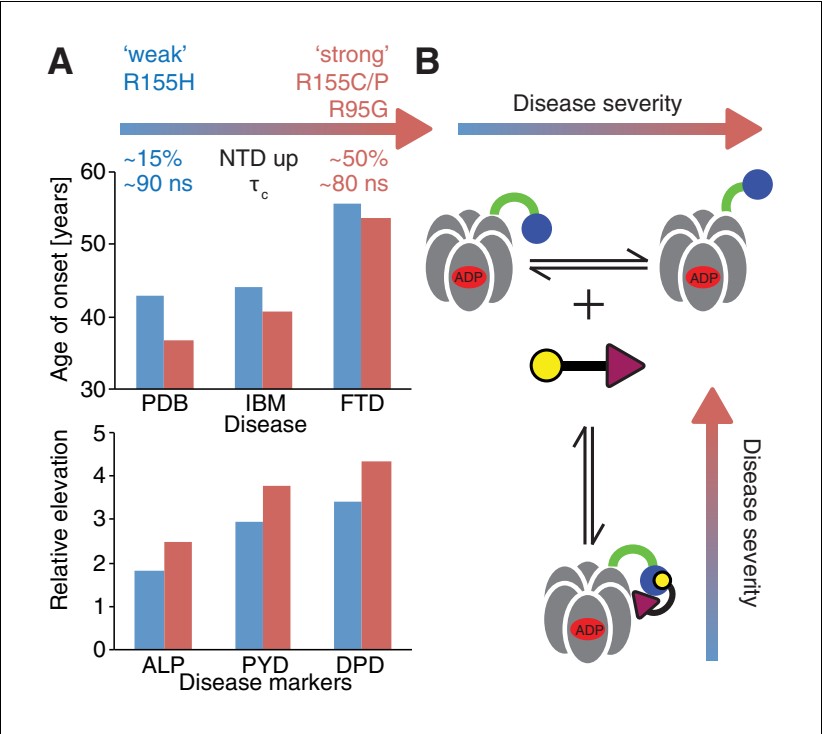

**Figure 10.** Correlation between disease severity and up/down equilibrium. (**A**) Disease (PDB=Paget disease of bone, IBM=inclusion body myopathy, FTD=frontotemporal dementia) onset and elevation of biochemical markers relative to normal (ALP=alkaline phosphatase, PYD=pyridinoline, DPD=deoxypyridinoline) (*Mehta et al., 2013*), is correlated with the extent of perturbation of the NTD up/down equilibrium. Data from R155H (R155C/P, R95G) patients are used for 'weak' ('strong') mutants; R155C is associated with a significantly earlier onset of symptoms compared to R155H and showed significantly reduced mean survival (*Mehta et al., 2013*). (**B**) Schematic of UBXD1-N binding to ND1Lp97-ADP.

The effects of IBMPFD disease mutations on p97 adaptor interactions, nucleotide binding, and nucleotide hydrolysis rates have been extensively investigated in a series of papers (*Fernández-Sáiz and Buchberger, 2010*; *Tang and Xia, 2013*; *Manno et al., 2010*; *Zhang et al., 2015*; *Bulfer et al., 2016*). Beyond the primary defect in UBXD1 binding (*Ritz et al., 2011*), altered cofactor interactions have been reported in a number of studies, including those of p47, p37, Ufd1-Npl4, E4B and ataxin-3 (41, 53, 54). Due to the NTD conformational equilibrium shift in disease mutants, changes in interactions are expected whenever adaptors bind differentially to the up and down conformations of the NTD, which are characteristic of ATP and ADP states of wt p97, respectively. Our experiments, that do not detect differences in p47 interactions with wt or disease mutants of p97, are consistent with biochemical studies of p47 binding to both wt and mutant variants of p97, for which affinities in the nM range are obtained (*Bulfer et al., 2016*). In these studies intact or even increased p47 binding of mutants has been detected (*Fernández-Sáiz and Buchberger, 2010*; *Bulfer et al., 2016*) and no differences in ATP hydrolysis kinetics between wt and disease mutant p97-p47 complexes in the limit of saturated binding have been observed (*Zhang et al., 2015*). Our NMR experiments are not sensitive to affinity differences for p47 and wt or mutant p97 since dissociation constants (nano-molar range) are below the threshold for which differences can be detected. They do establish, however, that the p97-p47 complex is formed analogously in wt and disease mutants (*Figure 8A*), and, unique to the NMR, that the skewed NTD conformational equilibrium found in disease mutants of p97-ADP is conserved in the p47 bound complex.

Several biochemical studies have reported altered nucleotide binding properties for disease mutants in terms of affinity and the amount of prebound nucleotide along with elevated ATPase activities (*Tang and Xia, 2013*). In general, the literature consensus is that disease mutations slightly

decrease the affinity of D1 for ADP (by factors between 2–4) but not for ATP, while increasing the ADP release rate from D1 by approximately 2-fold (*Tang et al., 2010*; *Tang and Xia, 2013*; *Bulfer et al., 2016*). We have observed no simple correlation between biophysical properties of various disease mutants and either perturbation of the NTD conformational equilibrium or empirical disease severity in patients. As an example, ATPase activities of disease mutants were reported to be normal in some (*Fernández-Sáiz and Buchberger, 2010*) and elevated in other studies (*Tang et al., 2010*; *Niwa et al., 2012*; *Tang and Xia, 2013*) but not in a manner that correlates with disease severity. Further, R155H is a weak disease mutation in patients (*Mehta et al., 2013*), with only a modest impact on the NTD up/down equilibrium, yet R155H p97 has been reported to have a particularly high ATPase activity and a low affinity for ADP (*Tang and Xia, 2013*). Thus, it is difficult to understand presently how altered nucleotide interactions can be the prevalent cause of disease, especially since it becomes hard to explain why only a subset of p97 functions would then be affected. In contrast, our results provide strong support for a model whereby perturbation of the up/down NTD equilibrium in the p97-ADP state due to mutation couples to disease by impairing p97 binding to the UBXD1 adaptor. Stronger perturbations lead to more acute disease (*Figure 10*). The binding of other adaptors, such as p47, that are involved in processes distinct from that of UBXD1, are much less affected, if at all, by the IBMPFD mutations.

In summary, this work provides a further demonstration of the utility of methyl-TROSY NMR in studies of high molecular weight complexes where dynamics play a critical role in both function and misfunction. While we report near-complete methyl group assignments for both ADP and ATP states of ND1Lp97 it is worth mentioning that the key conclusions about the disease mutants can be obtained without extensive assignment, by recording simple 2D $^{13}$C-$^1$H HMQC spectra of the IBMPFD mutant series, with quantification of the NTD up/down populations from the CSPs of only a few key reporter residues. Our NMR results paint a picture whereby the primary effect of disease causing mutations is a change in NTD dynamics accompanied by more subtle structural changes that propagate to regions distal from the sites of mutation. Perturbations to the up/down NTD equilibrium are observed in the form of CSPs that extend across the NTD/D1 interface and include the linker region, which in turn connects to the nucleotide-binding site (*Figure 9*). Residues showing large CSPs comprise a network that mediates communication between the NTD and D1 in wt p97 that extends to the central pore. Indeed, many of the 22 mutation sites reported to date in ND1Lp97 are localized to regions with large CSPs, suggesting that these loci serve as 'hot-spots' for propagation of conformational changes. Such sites may well serve as starting points for the rational design of drug molecules that modulate NTD-D1 interactions, potentially reversing the effects of the disease mutations or inhibiting the wt protein, a strategy currently explored in clinical cancer studies (*Anderson et al., 2015*).

## Materials and methods

### Plasmids and constructs

DNA encoding the p97 protein from *Homo sapiens/Mus musculus* (which are identical, Uniprot entry P55072 versus Q01853) was obtained from GenScript (Piscataway, NJ, USA) with an N-terminal 6xHis tag and TEV cleavage site, sub-cloned into the NdeI and XhoI sites of pET29b+ (Novagen, Madison, WI, USA). Point mutations and stop codons were introduced using Quikchange site-directed mutagenesis (Agilent, Santa Clara, CA, USA). The truncations used for this study were p97 ND1L 1–480 (*Song et al., 2003*) and p97 NTD 1–213 (*Isaacson et al., 2007*). DNA encoding adaptors, including full-length p47, p47 UBX (residues 282–371 [*Yuan et al., 2001*]) and UBXD1-N (residues 1–133 [*Trusch et al., 2015*]) from *Mus musculus* were also obtained from GenScript, with affinity tags, TEV cleavage sites and sub-cloning as for p97.

### Protein expression and purification

The isolated NTD of p97, ND1Lp97, full-length p97, p47 UBX, full-length p47 and UBXD1-N were overexpressed in *Escherichia coli* BL21(DE3) cells in minimal M9 $D_2O$ media supplemented with $^{15}NH_4Cl$ and $[^2H,^{12}C]$-glucose as the only nitrogen and carbon sources (Cambridge Isotope Laboratories (Tewksbury, MA, USA). Selective labeling with I-$\delta_1$-$[^{13}CH_3]$, V/L-$\gamma_1/\delta_1$(*proR*)-$[^{13}CH_3,^{12}CD_3]$ and M-$\epsilon_1[^{13}CH_3]$ was achieved as detailed previously (*Gelis et al., 2007*; *Gans et al., 2010*;

*Tugarinov et al., 2006*). Cells were induced at $OD_{600} \approx 0.8$ by addition of 1 mM IPTG and expression was performed for ca. 20 hr overnight at 18°C (25°C for UBXD1-N). p97 was purified using Ni-affinity chromatography and the affinity tag was cleaved using TEV protease. The cleaved proteins were concentrated using an Amicon Ultra-15 100 kDa molecular weight cutoff filter (Millipore, Etobicoke, ON, Canada) and nucleotide removed using apyrase (New England Biolabs, Ipswich, MA, USA, 2 units per NMR sample) for at least 2 hr (room temperature). The apo protein was further purified on a HiLoad 16/60 Superdex 200 gel filtration column (GE Healthcare, Pittsburgh, PA, USA). Buffer compositions were adapted from (*Chou et al., 2014*). BME (2 mM) and DTT (2 mM) were used as reducing agents during Ni-affinity chromatography and TEV cleavage, nucleotide digestion, gel filtration and for storage (DTT), respectively. Purification schemes for full-length p47 and p47 UBX were analogous to that used for p97, without reducing agent as there are no Cys residues in p47. In order to avoid proteolysis of full-length p47, protease inhibitor cocktail tablets (Roche, Basel, Switzerland) were added during all steps of purification except during TEV cleavage. Full-length p47 eluted as a trimer on a HiLoad 16/60 Superdex 200 gel filtration column and p47 UBX as a monomer on a HiLoad 16/60 Superdex 75 gel filtration column.

Purification of UBXD1-N was achieved with cell lysis and Ni-affinity chromatography under denaturing conditions (6 M guanidine), followed by TEV cleavage and size exclusion chromatography using a HiLoad 16/60 Superdex 75 column under native conditions.

Protein concentrations were determined based on absorbance at 280 nm for p97 and full-length p47. Neither p47 UBX nor UBXD1-N contains W or Y residues; protein concentrations for these constructs were thus determined via quantitative amino acid analysis. Proteins were subsequently exchanged into the appropriate NMR buffer or frozen at −80°C after addition of 5% glycerol for future use.

## NMR sample preparation

Protein samples of the isolated NTD were generated via exchange (10 kDa MWCO filters) into a buffer comprising 25 mM HEPES pH 7.5, 50 mM NaCl, 10 mM MgCl$_2$, 5 mM TCEP. Solvents that were either 90%/10% H$_2$O/D$_2$O ($^{15}$N-$^1$H based experiments) or 100% D$_2$O ($^{13}$C-$^1$H) were used along with protein concentrations up to 1.5 mM.

Optimal buffer conditions for full-length p97 and ND1Lp97 were identified via screening salt and nucleotide concentrations using dynamic scanning fluorimetry (using Sypro Orange (Sigma) as a reporter dye) for maximal melting temperature. ND1Lp97 samples were prepared for NMR via buffer exchange (100 kDa MWCO filters) into D$_2$O buffer containing 25 mM HEPES, pH 7.5(7.0), 25 (50) mM NaCl, 5 mM ATPγS(10 mM ADP), 4(0) mM MgCl$_2$, 5 mM TCEP for ND1Lp97-ATPγS (ND1Lp97-ADP) at protein concentrations between 100 μM (for assignment by mutagenesis), 600 μM (dynamics data) and 1 mM (NOESY based assignment). To limit ATP hydrolysis, ATPγS, a commercially available non-hydrolyzable analogue of ATP (Sigma/Roche), was employed in combination with the Walker B mutant of p97 (*Wendler et al., 2012*), in which the glutamate residue competent for ATP hydrolysis (E305Q) is replaced by glutamine. Samples were stable at 50°C for ~3 days in the ATPγS state, limited by ATPγS hydrolysis, and for about one week in the ADP form. It is worth emphasizing that under the conditions of our NMR experiments samples are fully nucleotide bound. For example, reported $K_d$ values for ADP binding to wt and disease mutants range from 0.1 μM – 5 μM, with the lower affinities associated with the disease mutants (*Tang et al., 2010*; *Tang and Xia, 2013*). Interestingly there is not a correlation between lower ADP affinity and disease severity as $K_d$(wt) = 0.88 ± 0.18 μM < $K_d$(R95G) = 2.3 ± 0.1 μM < $K_d$(R155H) = 4.3 ± 0.5 μM. Assuming $K_d$ = 5 μM, a concentration of p97 monomer of 600 μM (highest used) and [ADP]= 10 mM the amount of apo-p97 would be on the order of 0.01% of the nucleotide bound species.

NMR sample conditions for studies involving adaptors were identical to those used for p97. Concentrations (monomer) used for binding experiments were as follows: UBXD1-N 150 μM, wt/R95G/R155H/R155P/N387H ND1Lp97-ADP 50 μM; full-length p47 300 μM, wt/R95G NDL1p97-ADP 250 μM; p47 UBX 300 μM, ND1Lp97-ADP 200 μM. All p97 binding partners were perdeuterated. Our NMR studies necessitate monomer protein concentrations in the tens-hundreds of μM range, depending on the application. Interestingly, as protein concentrations can reach 200 mg/mL (*Ellis, 2001*) or ~5 mM in the cytosol, with p97 accounting for an estimated 1% of cytosolic proteins, the NMR experiments are carried out under conditions that resemble physiological. Of note, samples of the ND1Lp97-ADP/UBXD1-N complex were stable for ca. 5 hr at 50°C, at which point

precipitation of UBXD1-N leads to increasing amounts of unbound p97. In contrast, the tightly bound p47-p97 complexes were stable for at least 12 hr.

## Cross-linking

Covalent cross-linking of NTD to D1 in ND1Lp97 was achieved by mutating R155 (NTD) and N387 (D1) to Cys (*Niwa et al., 2012*). The resulting protein was purified as described above using reducing conditions and dialyzed into non-reducing buffer for 3 days at 4°C prior to NMR experiments. For R95G ND1Lp97 oxidation was more difficult than for the wt protein and required the addition of the oxidizing agent Cu phenanthroline (*McIntosh and Freedman, 1980*; *Barthelme et al., 2014*). The increased difficulty in oxidation of the R95G sample is consistent with the fact that the NTD equilibrium in the mutant is much further in the up direction so that cross-linking, which only occurs from the NTD down position, is less efficient. The completion of the reaction was monitored by NMR spectroscopy via chemical shift changes of the *proR* methyl group of V154, adjacent to C155.

## Data acquisition and analysis

NMR experiments were performed at field strengths of 14.0 T and 18.8 T using either Varian or Bruker spectrometers. With the exception of several 3D data sets and the adaptor data all experiments at 18.8 T were measured on a system with a room temperature probe; the remaining experiments were obtained using cryogenically cooled probes. Experiments were recorded at 37°C on the isolated NTD and at 50°C for all samples involving ND1Lp97 and full-length p97. Spectra were processed using the NMRPipe suite of programs (*Delaglio et al., 1995*) with chemical shift assignments obtained with the aid of the CcpNMR (*Vranken et al., 2005*) program. Peak fitting was performed using FuDA (http://pound.med.utoronto.ca/~flemming/fuda/), with remaining data analysis achieved via home-built scripts executed in MATLAB.

## NMR assignments

### Assignment of p97 NTD

The NTD domain of p97 is stable and monomeric (24 kDa) and sequential backbone resonance assignments have been obtained before (*Isaacson et al., 2007*). These were kindly provided to the authors by Professor Rivka Isaacson (King's College, London). Resonance assignments were repeated using standard TROSY-based (*Pervushin et al., 1997*) triple resonance experiments (*Sattler and Schleucher, 1999*) (HNCO, HNCACO, HNCACB, CBCA(CO)NH) on a uniformly [$^{13}$C,$^{15}$N,$^2$H] labeled sample and subsequently extended to the methyl sidechains using (H)C(CO)NH-TOCSY, H(CCO)NH-TOCSY schemes (*Gardner et al., 1996*) ([$^{13}$C,$^{15}$N, ~70% $^2$H] labeled sample). As a final step, assignments were confirmed via analysis of a 3D $^{13}$C-edited NOESY data set (200 ms mixing time) that was modified to include the methyl-TROSY approach (sequence available upon request). The experiment was recorded as $^{13}$C($t_1$) NOE $^{13}$C($t_2$),$^1$H($t_3$) where $t_i$ is a chemical shift acquisition period. Assigned NOEs were cross-validated based on the X-ray structure of the domain (*Hänzelmann et al., 2011*) (pdb ID 3QQ7). Near complete assignments of methyl groups of ILVM residues were obtained (56/59), while those that could not be assigned (L17, I32, M158) did not give rise to peaks in spectra. Stereospecific assignment of V and L methyl groups was achieved as reported by Neri et al. (*Neri et al., 1989*).

### Assignment of methyl groups of ND1Lp97

NTD assignments were subsequently transferred to spectra of ND1Lp97 (6x53 kDa) in ADP and ATPγS nucleotide states via an NOE-based approach using 3D $^{13}$C-edited NOESY data sets (*Sprangers and Kay, 2007*). Methyl chemical shifts for residues in the D1 domain were obtained via a mutagenesis strategy (*Sprangers and Kay, 2007*), whereby assignments are generated by mutating methyl containing residues and comparing the resulting spectra with the corresponding data set recorded on the wt protein. Assignments were extended via analysis of NOE data sets where measured distances between proximal methyl groups in solution, as estimated qualitatively by NOE cross-peak intensities, were compared with those predicted on the basis of p97 X-ray structures (ADP state pdb ID 1E32 [*Zhang et al., 2000*]; ATP state pdb 4KO8 [*Tang and Xia, 2013*]). Spectra of 40 point-mutants of residues distributed throughout D1 were recorded in both ADP and ATPγS states, with remaining residues assigned based on NOEs (150 ms mixing time) to these 40

'anchoring' positions. In the ADP state, 128/132 (97%) of all methyl probes could be assigned, including 54/54 in the NTD, 4/4 in the linker region and 70/74 in the D1 domain. The total of 132 methyl groups is based on including only 1 methyl for each V/L residue as labeling of these residues was *proR* (*Gans et al., 2010*). Assignments for 104/132 (79%) of the methyl groups in ND1Lp97-ATPγS were obtained (54/54, 4/4 and 46/74 in the NTD, linker and D1, respectively). The point mutants utilized for assignment were chosen based on prediction of mutant stability from △△G calculations (*Kellogg et al., 2011*) and sequence homology alignment.

## Dynamics measurements

### NTD correlation time

$^{15}$N $R_1$ and $R_{1\rho}$ relaxation rates and steady state $^{15}$N-{$^1$H} NOE values (*Farrow et al., 1994*) for the isolated NTD were acquired at 600 MHz, 37 °C using relaxation delays of 10, 110, 240, 380, 550, 750, 1000 ms ($R_1$) and 2, 7, 14, 22, 31, 42, 55 ms ($R_{1\rho}$) along with a 1.8 kHz spin-lock field ($\nu_1$). $R_{1\rho}$ values were converted to $R_2$ rates via the relation, $R_{1\rho} = R_2 \sin^2(\theta) + R_1 \cos^2(\theta)$, where $\tan(\theta) = \nu_1 / \triangle$, and $\triangle$ is the $^{15}$N chemical shift offset from the carrier. Per-residue values of $\tau_c, S^2$ and $\tau_e$ were obtained via a spectral density mapping approach (*Farrow et al., 1995*), where $\tau_c$ is the assumed isotropic molecular tumbling time, $S^2$ is the square of an order parameter that is related to the amplitude of the amide bond vector motion and $\tau_e$ is the correlation time for rapid bond vector dynamics. Briefly, values of the spectral density function at three frequencies were obtained for each residue and these were fit to extract ($\tau, S^2, \tau_e$). A value of $\tau_c = 13.6 \pm 0.9$ ns was obtained, where the error is given as 1 standard deviation of the range of values, using only those residues in structured regions of the protein. This value is in keeping with expectations for a 24 kDa protein, 37°C (*Jarymowycz and Stone, 2006*). The value of $\tau_c$ so obtained was subsequently used to calculate methyl axis order parameters, as described below.

### Measurement of isolated NTD domain methyl $S_{axis}^2 \tau_c$ values

Methyl $^1$H spin relaxation data sets were recorded (37°C) as described in (*Sun et al., 2011*) using an approach that quantifies the time dependencies of sums and differences of magnetization that give rise to $^1$H single- ($I_{SQ}$) and triple- ($I_{TQ}$) quantum coherences, respectively. Relaxation delay values $T$ of 2, 7, 12, 17, 22, 27, 32, 37, 42, 50 ms were recorded in an interleaved manner and intensities of methyl cross peaks fit according to

$$\left| \frac{I_{SQ}}{I_{TQ}} \right| = \frac{0.75 \eta \tanh(T\sqrt{\eta^2 + \delta^2})}{\sqrt{\eta^2 + \delta^2} - \delta \tanh(T\sqrt{\eta^2 + \delta^2})} \tag{2}$$

with $S_{axis}^2 \tau_c$ obtained via

$$\eta \cong \frac{9}{10} \left( \frac{\mu_0}{4\pi} \right)^2 \left[ P_2 \left( \cos\theta_{axis,HH} \right) \right]^2 \frac{S_{axis}^2 \gamma_H^4 \hbar^2 \tau_C}{r_{HH}^6} \tag{3}$$

In *Equation 3* $P_2(x) = (3x^2 - 1)/2$ with $\theta_{axis,HH} = 90°$ the angle between the methyl threefold axis and the vector connecting a pair of methyl protons, $S_{axis}^2$ is the square of an order parameter quantifying the amplitude of motion of the methyl threefold symmetry axis, $\hbar$ is Planck's constant divided by $2\pi$, $\gamma_H$ is the gyromagnetic ratio of a proton spin, and $r_{HH}$ is the distance between pairs of methyl protons (1.813 Å). The parameter $\delta$ in *Equation 2* accounts for the proton density around each methyl group. Values of $S_{axis}^2$ were obtained for each methyl group from the quantified $S_{axis}^2 \tau_c$ values, assuming $\tau_c = 13.6$ ns as obtained above from $^{15}$N spin relaxation experiments. We did not attempt to take into account any motional anisotropy, as the orientation of methyl groups from X-ray structures must be considered to be approximate, especially in cases where sidechains are dynamic.

### Determination of $\tau_c$ for ND1Lp97

Methyl $^1$H spin relaxation data sets, as described above, were recorded on samples of ND1Lp97 (for wt and mutant proteins in ADP, ATPγS states) using relaxation delay values of 0.5, 1, 2, 3, 4, 5, 6, 7, 8, 10, 12, 15, 18 ms. Twenty NTD residues were considered for analysis based on minimal chemical shift changes between corresponding methyl groups in the isolated NTD and in the NTD of

ND1Lp97. This ensures that individual NTD methyl group dynamics are little affected by the surrounding D1L in ND1Lp97 constructs. Thus, $S^2_{axis}$ values measured for isolated NTD would be expected to be similar for the corresponding NTD methyl groups in ND1Lp97. $S^2_{axis}$ values obtained for residues in the isolated NTD (37°C) were extrapolated to 50°C by assuming a temperature-dependence as described by Wand and coworkers (*Song et al., 2007*). These extrapolated order parameters were then used to obtain a global estimate of the $\tau_c$ for the NTD of ND1Lp97, *Figure 6—figure supplement 1*, via fitting measured $S^2_{axis}\tau_c$ values for ND1Lp97 using values for $S^2_{axis}$ obtained as described above. A $\tau_c$ value for the hexameric barrel formed by the D1 domains was obtained from maximum $S^2_{axis}\tau_c$ values measured for methyl groups of D1, as described previously (*Sprangers and Kay, 2007*). The value of 120 ns obtained in this manner is in good agreement with that obtained for the half-proteasome, $\alpha_7\alpha_7$, that is of a similar size (360 kDa, 50°C) to NDL1p97 (*Sprangers and Kay, 2007*). Interestingly, HYDRONMR (*García de la Torre et al., 2000*) predictions for $\tau_c$ of ND1Lp97 were 150–200 ns, depending on which X-ray structure was used in the calculations (1E32 [*Zhang et al., 2000*], 3HU1 [*Niwa et al., 2012*], 4KOD [*Tang and Xia, 2013*]).

## Analysis of UBXD1-N adaptor titration

A previous study has established that UBXD1-N has two p97 binding regions (*Trusch et al., 2015*). These include a well-characterized VIM domain that forms a single ~10 residue helix upon binding to a groove between NTD sub-domains (*Stapf et al., 2011*; *Hänzelmann and Schindelin, 2011*) and an H1/H2 helical domain that has been shown through biochemical studies to fix the NTDs of wt p97-ADP in the down position (referred in our work as the locked, down position). The results presented in *Figure 7* of the main text establish a two-pronged binding mode with NTD residues at the canonical UBX/VIM binding interface reporting on VIM binding (for example V38, V68, V108, see *Figure 7*, *Figure 7—figure supplement 2D,E*), with residues such as I146 and I189 reporting on H1/H2 binding.

The CSPs of V68 from the addition of UBXD1-N to R95G ND1Lp97-ADP has been used to estimate a $K_d$ for the VIM ND1Lp97-ADP interaction. This mutant allows the deconvolution of VIM and H1/H2 binding because the latter interaction does not occur (CSPs are not observed for I146/I189 of the R95G mutant). The fraction of bound VIM for [NTD]$_T$ = 6x[ND1Lp97-ADP] = 50 μM, [UBXD1-N]$_T$ = 150 μM considered here, where [NTD]$_T$ and [UBXD1-N]$_T$ are the total (T) concentrations of NTD and UBXD1-N, respectively, can be calculated from the shift ($\delta_i$) of V68 (black peak, *Figure 7D* inset) and the difference in chemical shifts between free and bound states, $\delta_B - \delta_F$, via,

$$f_B = \frac{\delta_i - \delta_F}{\delta_B - \delta_F}. \tag{4}$$

Note that the affinity of VIM for the R95G mutant is relatively weak so that VIM binding was not saturated with the 3:1 ligand to p97 (monomer) ratio used and calculations establish that a very high excess of UBXD1-N would be required for saturation. The value of $\delta_B$ used in the calculation was, therefore, obtained from the wt protein with a 3:1 ratio (same $\delta_B$ with a 5:1 ratio) since the wt protein has a higher net affinity for UBXD1-N (compare grey and black peaks in the inset to *Figure 7D* for V68). In this manner $f_B = [B]/([F] + [B])$ for the R95G mutant was obtained (~0.4) from which a $K_d$ of ~200 μM was calculated, 50°C. The poor sample stability (aggregation of UBXD1-N after ca. 5 hr) precluded a typical titration, as is routinely done for stable samples. The obtained $K_d$ is a microscopic value for the interaction of the VIM domain of UBXD1-N with the NTD of ND1Lp97-ADP.

We have also calculated an effective macroscopic $K_d (K_{d,macro})$ for the binding of UBXD1-N to wt ND1Lp97-ADP. CSPs for residues that report on the VIM - NTD interaction are influenced by a number of different pathways, including direct binding as well as binding that follows the H1/H2 domain-NTD association (see *Scheme 1* and discussion below). In this case $K_{d,macro}$ = 22 ± 2 μM, 50°C, is obtained from analysis of chemical shift titration profiles for V68, L140, I175, I182 (*Figure 7*, *Figure 7—figure supplement 3*). Because of the instability of UBXD1-N at 50°C, each concentration of ligand required a new sample, limiting the number of titration points. The corresponding $K_{d,macro}$ value for the moderate disease mutant, R155H, has been obtained from a 1 point titration, as for R95G above, yielding a value of ~80 μM, 50°C.

The CSPs for I146/I189 of the different disease mutants provide valuable information about how the individual IBMPFD mutations affect binding of the H1/H2 motif, as illustrated below. In the case

of wt ND1Lp97-ADP small shifts of I146/I189 occur (as well as for other residues, see *Figure 7—figure supplement 2B,C*) that are similar to those observed upon locking the down state via disulfide bond formation. For moderate disease mutants such as R155H and N387H where $p_U \sim$ 15% in the free state, the observed I146/I189 CSPs correspond to a slight shift in $p_U$ to ~10% that reflects binding of H1/H2 (although the calculated shift is larger for I146 than for I189 based on CSPs), with no CSPs, and hence effectively no H1/H2 binding for the R95G mutant. We consider below a simple model in which the disease mutants are assumed to have little effect on the *microscopic* binding of the VIM domain to the NTD (equilibrium defined by $K_2$ below) but where H1/H2 binding becomes increasingly impaired with severity of mutation. In this manner UBXD1 binding locks the NTD in the down state for wt ND1Lp97-ADP, while for moderate mutants (R155H, *Figure 7C*; N387H, *Figure 7—figure supplement 2D*) an incomplete fraction of the down state is locked (both VIM and H1/H2 bound). In contrast, the binding affinity for H1/H2 becomes sufficiently weak in the case of strong disease mutants (R95G, *Figure 7D*; R155P, *Figure 7—figure supplement 2E*) so that the up/down equilibrium cannot be shifted to the down, locked state at all. The binding model of *Scheme 1* can be used to quantify this further, providing a picture that is consistent with experiment.

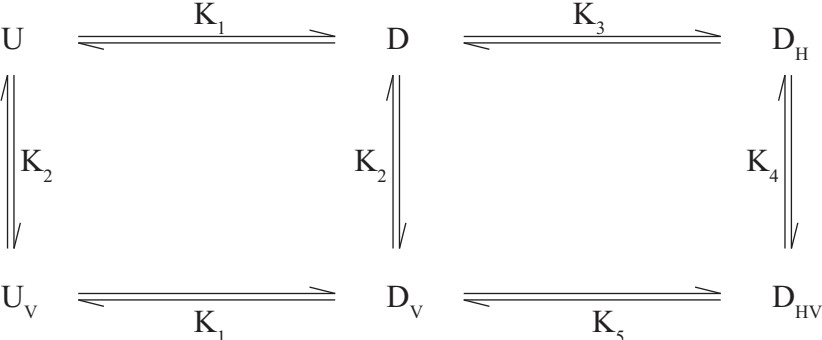

**Scheme 1.**

Here $U(D)$ and $U_V(D_V)$ correspond to NTD in the up(down) states, either free ($U, D$) or bound ($U_V, D_V$) to the VIM motif of UBXD1-N, and $D_H$, $D_{HV}$ denote the down state that is bound to UBXD1-N via the H1/H2 motif ($D_H$) or via both VIM and H1/H2 motifs ($D_{HV}$). The $K_i$ values are given by: $K_1 = [U]/[D] = [U_V]/[D_V]$, $K_2 = [U][L]/[U_V] = [D][L]/[D_V]$, $K_3 = [D][L]/[D_H]$, $K_4 = [D_H]/[D_{HV}]$, $K_5 = [D_V]/[D_HV]$, L=UBXD1-N, from which it follows that $K_4/K_5 = K_2/K_3 = 1/\gamma$. This model assumes (i) that each NTD acts independently, (ii) that the up/down equilibrium is unaffected by binding V, (iii) that binding of H can only occur for an NTD down state and (iv) equivalent binding affinities of V to p97 when NTDs are in the up or down states. Condition (iii) follows from the fact that it is only possible for H1/H2 to bind to the NTD-D1 interface when the NTDs are close to the interface (*i.e.*, down). It is worth noting that $K_{d,macro}$ as measured in the UBXD1-N titration of wt ND1Lp97-ADP and reported by the chemical shifts of methyl probes that are sensitive to VIM binding (such as V68) is given by the relation

$$K_{d,macro} = \frac{([D] + [U] + [D_H]) \cdot [L]}{([U_V] + [D_V] + [D_{HV}])} = \frac{K_2(1 + K_1) + \frac{[L]}{\gamma}}{1 + K_1 + \frac{1}{K_5}}. \tag{5}$$

It is straightforward to show that in *Scheme 1*, above, $[L]$ is given by the positive root of the equation $A[L]^2 + B[L] + C = 0$ where

$$
\begin{aligned}
A &= \alpha\left(\frac{1}{K_2} + \frac{1}{K_1 K_2} + \frac{1}{K_1 K_3} + \frac{1}{K_1 K_3 K_4}\right), \ \alpha = \left(\frac{K_1}{1 + K_1}\right)\\
B &= A(P_T - L_T) + 1\\
C &= -L_T
\end{aligned}
\tag{6}
$$

and $P_T$, $L_T$ are the total protein and ligand concentrations (50 μM and 150 μM in the present study). For wt, moderate (R155H and N387H) and severe (R95G, R155P) mutants, $K_1 << 1$, $K_1 \sim$ 0.15, $K_1 \sim$

1, respectively (*Figure 6*). A value of $K_2 \sim 200$ µM has been measured, as described above, and although a value for $K_3$ is not available from our studies (too weak) it has been measured to be approximately γ=25 fold larger than $K_2$ in studies of an isolated NTD (*Trusch et al., 2015*). We have carried out simulations assuming $K_5$=0.1 (wt), $K_5$=0.5 (R155H, N387H) and $K_5$=10 (R95G, R155P) (*i. e.*, locking of the NTD becomes progressively more difficult with severity of mutation), with $K_3$ and $K_4$ obtained from $K_2 = K_3/\gamma$ and $K_4 = K_5/\gamma$, respectively, assuming γ=25 for wt and γ increasing for the mutants in proportion to $K_5$(mutant)/$K_5$(wt) so that $K_2$ and $K_4$ are fixed (microscopic binding of VIM unperturbed by mutation). Thus, there is only a single free parameter in scheme 1 ($K_5$). Simulations (for $P_T$= 50 µM, $L_T$= 150 µM, as in experiments) show that in the case of wt ND1Lp97-ADP the down conformation is pushed to the locked-form upon binding UBXD1-N, as expected from experiment. For moderate disease mutants a considerable fraction of the down state is locked (~40%). In addition, the up/down equilibrium becomes more skewed towards down (~90%, $p_U \sim$ 10%), as observed experimentally (*Figure 7C*, *Figure 7—figure supplement 2D*). In contrast, in the case of severe mutations, the populations of the up and down protein states remain essentially unchanged (as observed experimentally), with only a few percent of the locked, down state produced. Notably, the results change little for γ=5 and even when $L_T$ is increased to 300 µM (6-fold over $P_T$) there is still little locked, down state produced. Values of $K_{d,macro}$ obtained from the model are ~ 20 µM, 80 µM and 200 µM for wt, moderate and severe disease mutants, respectively, consistent with experiment. Indeed, in the case of wt ND1Lp97-ADP where $1/K_5 \gg 1 + K_1, K_1 \ll, 1 \ K_{d,macro} \sim K_5 K_2$, while for severe disease mutants, $1/K_5 \ll 1 + K_1, \gamma$ is very large and $K_{d,macro} \sim K_2$ (*Equation 5*). The proposed model can explain the CSPs of *Figure 7* and, importantly, it shows that $K_5$ must increase with disease severity because keeping $K_5$ constant cannot explain the observation that the up/down equilibrium is shifted somewhat for moderate mutants but not at all for severe mutations with addition of UBXD1-N.

## Acknowledgement

The authors thank Drs. R Rosenzweig, A Sekhar, A Velyvis, and R Vernon for expert advice, Dr. R Muhandiram for excellent technical support and Dr. R Isaacson (King's College) for assignments of the isolated p97 NTD. AKS acknowledges funding from the Swiss National Science Foundation and the European Molecular Biology Organization. This work was supported by a grant from the Canadian Institutes of Health Research. LEK hold a Canada Research Chair in Biochemistry.

## Additional information

### Funding

| Funder | Grant reference number | Author |
| --- | --- | --- |
| European Molecular Biology Organization | ALTF 100-2013 | Anne K Schuetz |
| Swiss National Science Foundation | P2EZP2_148754 | Anne K Schuetz |
| Canadian Institutes of Health Research | RN203972 - 310401 | Lewis E Kay |

The funders had no role in study design, data collection and interpretation, or the decision to submit the work for publication.

### Author contributions

AKS, LEK, Conceived the study, Produced samples, Obtained NMR spectra and analyzed the data, Wrote and edited the paper, Involved in all aspects of the work

### Author ORCIDs

Lewis E Kay, http://orcid.org/0000-0002-4054-4083

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
