## [Decision Letter]

Thank you for submitting your article "A Dynamic Molecular Basis for Malfunction in Disease Mutants of p97/VCP" for consideration by *eLife*. Your article has been favorably evaluated by Richard Aldrich as the Senior Editor and three reviewers, one of whom, Volker Dötsch (Reviewer #1), is a member of our Board of Reviewing Editors.

The reviewers have discussed the reviews with one another and the Reviewing Editor has drafted this decision to help you prepare a revised submission.

Summary:

The manuscript by Schuetz and Kay describes the effect of disease mutations on the conformational dynamics of the AAA-ATPase p97. This molecular machine is a subject of much research because of its central role in proteolysis that occurs during the ubiquitin-proteasome and autophagy pathways. Inhibition of p97 is the subject of much research, and there are compounds in clinical trials for treatment of cancer, making this topic of broad interest.

Recent crystallographic and cryoEM studies document a nucleotide dependent conformational change in p97. This AAA-ATPase consists of two hexameric rings that are coaxially stacked. One ring contains the N-terminal domain (NTD) and first ATPase domain (D1) which is linked to the other ring containing a second ATPase domain (D2). In the wild-type ATPase, the NTD undergoes conformational changes from a co-planer (or down) to up position upon exchange of ADP with ATP in the D1 domain. In a tour de force effort, Schuetz and Kay provide evidence that there is a fast, preexisting equilibrium in this down/up transition of the NTD, with disease mutations shifting the equilibrium from the ADP-down state to an ADP-up state. The ADP-up state may be considered aberrant because the authors observe a correlation between the fraction of the ADP-up state for various patient derived mutations and severity of disease. Moreover the authors provide evidence that binding of p97 to the UBXD family is sensitive to the up/down equilibrium whereas other adaptors, such as p47, are not affected.

Essential revisions:

1) There are 22 disease causing mutation sites known, of which 6 were selected for this study. What was the criterion for selecting exactly these 6 and can the other sites also be explained with the results obtained with the 6 sites investigated?

2) The authors state that the dissociation constants for p97 and UBXD1-N studied are in the low nanomolar range. Is this also true for the patient derived mutants? If this information is known from prior studies, it would be useful to tabulate it in the supplement. Without this information, it is difficult to interpret Figure 4. i.e. – are the CSP effects in mutants from a reduced fraction bound or is p97 fully bound by UBXD1-N but in a different mode/orientation for mutations studied?

3) Related, it seems that the binding of H1/H2 of UBXD1-N would be thermodynamically (and structurally) coupled to formation of the down state since the interface of D1/NTD might be a binding site for this region of UBXD1. If p97 is not completely bound by UBXD1-N for some of the mutants tested in Figure 4, then increasing the concentration of UBXD1-N in the NMR studies might fully restore the down state of the NTD, which would reflect a more complete return of chemical shifts to those of ADP state of WT. Have the authors tested this?

4) The authors may wish to comment on why some cross-peaks of the p97D1L construct are completely missing in the context of full-length p97 (Figure 1). Is this from increased tumbling or from additional ms-μsec dynamics in full-length p97?

5) Does the affinity for ADP change for the mutants studied. This seems plausible given the allostery network the authors observe in this system (Figure 2). Though it seems likely the nucleotide binding site is fully occupied at concentrations of p97 and nucleotide used for NMR, the authors ought to address this point.

6) Why do of the chemical shift change trajectories deviate slightly from linearity?

7) The authors speculate about allosteric networks that are responsible for the different orientation of the N-terminal domain. Can this be further analyzed and be explained with concrete interactions between amino acids that are changed in the mutants?

---

## [Author Response]

*[…] Essential revisions:*

*1) There are 22 disease causing mutation sites known, of which 6 were selected for this study. What was the criterion for selecting exactly these 6 and can the other sites also be explained with the results obtained with the 6 sites investigated?*

We have modified the text to include the following:

“The selected sites span the complete NTD-D1 interface (7800 Å^2^) and NTD-D1 linker region over which the disease mutations are localized. Moreover, the sites chosen for analysis are very close in space to the majority of the other disease causing mutations and thus are likely to be representative of these as well, Table 1.”

We have also included an additional section to a figure (Figure 2) that includes a summary both of the mutants used and other disease mutations that are proximal to each of the mutations that are considered in the present manuscript. This highlights the excellent coverage with the mutations considered. Moreover, we have also included an additional site in our analysis, A232, that is involved in moderate disease.

*2) The authors state that the dissociation constants for p97 and UBXD1-N studied are in the low nanomolar range. Is this also true for the patient derived mutants? If this information is known from prior studies, it would be useful to tabulate it in the supplement. Without this information, it is difficult to interpret Figure 4. i.e. are the CSP effects in mutants from a reduced fraction bound or is p97 fully bound by UBXD1-N but in a different mode/orientation for mutations studied?*

The reviewer is not correct regarding this statement. The dissociation constants for wt-p97 and UBXD1-N are in the µM range. In contrast, the dissociation constants for p97 and p47 (a different adaptor) are indeed in the nM range and perhaps this is what was confusing. We have done additional experiments to verify the affinities of UBXD1-N for wt and disease mutants and have added the following to the text:

“Larger shifts are observed for V68 and I175 that report on VIM binding, from which a *K_d_*value of 22 ± 2 µM for the VIM-NTD interaction is calculated (see Materials and methods Figure 7—figure supplement 3). This value is in good agreement with affinities measured for the binding of full-length wt p97 to full length UBXD1, 3.5 µM (36), and for the binding of the isolated NTD with UBXD1-N, 9 µM (35), especially considering that the latter pair of studies were carried out at 25^°^C in comparison to 50^°^C for the NMR results reported here.”

“Finally, it is worth emphasizing that the similar trajectories of CSPs as a function of added UBXD1-N observed for wt and all disease mutants (note the collinearity of dashed and solid arrows) provides strong evidence that the mechanism of binding is identical in all cases, with the differences in the extent of shift changes reflecting differing affinities, as calculated in the present study.”

In addition we have added an extra supplementary figure (Figure 7—figure supplement 3) illustrating the titration of wt p97-ADP with UBXD1-N. We must emphasize, as we have now done in the text, that the similar trajectories of CSPs observed for wt and disease mutants argues very strongly in favor of the same binding mechanism, with the differences in shifts reflecting differences in affinities. Finally, as discussed in the subsection “Implications of mutations on UBXD1 binding”, in addition to measuring the *K_d_*for the UBXD1-N / wt ND1Lp97-ADP interaction (22 µM), we have also obtained the *K_d_* for the weak disease mutant R115H (80 µM) along with the *K_d_* for the strong disease mutant R95G (200 µM).

*3) Related, it seems that the binding of H1/H2 of UBXD1-N would be thermodynamically (and structurally) coupled to formation of the down state since the interface of D1/NTD might be a binding site for this region of UBXD1. If p97 is not completely bound by UBXD1-N for some of the mutants tested in Figure 4, then increasing the concentration of UBXD1-N in the NMR studies might fully restore the down state of the NTD, which would reflect a more complete return of chemical shifts to those of ADP state of WT. Have the authors tested this?*

The reviewer is indeed correct in that, in general, it should be possible to push the equilibrium through the addition of increasing amounts of UBXD1-N. As discussed in the text, binding is two-pronged and the microscopic affinity for prong 1 (VIM domain) is low (200 µM) but higher than for the second prong (H1/H2). We have carried out studies at a 3:1 excess of UBXD1-N over p97 (protomer) and would require a significantly higher excess, especially for the R95G mutant of interest (where no binding of prong 2 is observed), to push the equilibrium substantially given these affinities. Because the 3:1 experiment required 3L of D_2_O growth to obtain enough of UBXD1-N we are reluctant to pursue this further. However, we present a binding model where we show that even at a 6:1 excess there would essentially be no measurable change to the locked, down conformation in the severe disease mutants (see ‘Analysis of UBXD1-N adaptor titration’ in Materials and methods).

*4) The authors may wish to comment on why some cross-peaks of the p97D1L construct are completely missing in the context of full-length p97 (Figure 1). Is this from increased tumbling or from additional ms-μsec dynamics in full-length p97?*

We have modified the text:

“Well-resolved resonances in ^13^C-^1^H HMQC spectra of ILVM-^13^CH_3_-ND1Lp97 and full length p97 (6*89 kDa) labeled in the same manner are superimposable (Figure 3—figure supplement 1), establishing that ND1Lp97 is a good model system for structural studies. Notably, some peaks in spectra of full-length p97 are missing in the comparison that reflects the slower tumbling of the larger complex, leading to inferior spectra relative to the 320 kDa construct.”

*5) Does the affinity for ADP change for the mutants studied. This seems plausible given the allostery network the authors observe in this system (Figure 2). Though it seems likely the nucleotide binding site is fully occupied at concentrations of p97 and nucleotide used for NMR, the authors ought to address this point.*

Indeed, the affinities do change, but quite subtly. In our studies the nucleotide binding site is, however, fully occupied. We now state this in the revised manuscript:

“It is worth emphasizing that under the conditions of our NMR experiments samples are fully nucleotide bound. For example, reported *K_d_* values for ADP binding to wt and disease mutants range from 0.1 µM – 5 µM, with the lower affinities associated with the disease mutants (20, 50). Interestingly there is not a correlation between lower ADP affinity and disease severity as *K_d_*(wt) = 0.88 ± 0.18 µM < *K_d_*(R95G) = 2.3 ± 0.1 µM < *K_d_*(R155H) = 4.3 ± 0.5 µM. Assuming *K_d_* = 5 µM, a concentration of p97 monomer of 600 µM (highest used) and [ADP]= 10 mM the amount of apo-p97 would be on the order of 0.01% of the nucleotide bound species.”

*6) Why do of the chemical shift change trajectories deviate slightly from linearity?*

We address this as follows:

“Deviations from linearity, observed for a fraction of the probes, likely reflect small structural perturbations beyond the principal effect of the mutation (see below). In this context, it is noteworthy that in the general case non-linear changes in shifts can be observed even in the case of two-site exchange when the chemical shift time-scales for ^13^C and ^1^H nuclei are different, that can arise from large chemical shift differences for one nuclei and smaller changes for the other.”

*7) The authors speculate about allosteric networks that are responsible for the different orientation of the N-terminal domain. Can this be further analyzed and be explained with concrete interactions between amino acids that are changed in the mutants?*

In the revised manuscript we include a new section:

“A key strength of NMR in studies of proteins lies in the fact that the easiest parameter to measure – the chemical shift – is also amongst the most powerful in detecting subtle conformational changes of the sort that can give rise to allosteric pathways of communication, for example. […] Finally, the NTD conformational change is communicated to the adjacent protomer at the I27/L429 interface via pathway III and this network eventually connects to the same cluster of residues as in pathway II.”

Additionally, we have added a new main text figure (Figure 9) that highlights the allosteric paths as quantified by the chemical shift changes that we observe in the R95G disease mutant.